https://doi.org/10.5194/egusphere-2024-3677

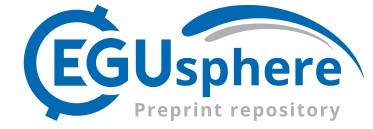

# Investigating the impact of subgrid-scale aerosol-cloud interaction on mesoscale meteorology prediction

Wenjie Zhang[1], Hong Wang[1], Xiaoye Zhang[1], Yue Peng[1], Zhaodong Liu[2], Deying Wang[1], Da Zhang[3], Chen Han[1], Yang Zhao[1], Junting Zhong[1], Wenxing Jia[1], Huiqiong Ning[1], Huizheng Che[1]

[1]State Key Laboratory of Severe Weather & Key Laboratory of Atmospheric Chemistry of CMA, Chinese Academy of Meteorological Sciences, Beijing, China
[2]Earth System Modeling and Prediction Centre, China Meteorological Administration, Beijing, China
[3]Institute of Energy, Environment and Economy, Tsinghua University, Beijing, China

*Correspondence to*: Hong Wang (wangh@cma.gov.cn)

**Abstract.** Aerosol-cloud interaction (ACI) significantly influences global and regional weather systems and is a critical focus in numerical weather prediction (NWP), but subgrid-scale ACI effects are often overlooked. Here, subgrid-scale ACI mechanism is implemented by explicitly treating cloud microphysics in KFeta convective scheme, which realizes real-time subgrid-scale size-resolved hygroscopic aerosol activation and cloud radiative feedback, in a mesoscale atmospheric chemistry model CMA_Meso5.1/CUACE to investigate its impacts on meteorology prediction in summer over central and eastern China.

Results show that incorporating subgrid-scale ACI refines cloud representation even in some grid-scale unsaturated areas and subsequently leads to attenuated surface downward shortwave radiation with regional mean bias (MB) decreasing by 23.1%. The increased cloud radiative forcing results in lower temperature and higher relative humidity (RH) at 2 m, helping to reduce regional MB by 40% and 18.1%. Temperature vertical structure and RH below ~900 hPa are improved accordingly due to cooling and humidifying. Subgrid-scale ACI further significantly enhances precipitation, especially at grid-scale, thus reducing

regional MB by 34.4%. The differences in subgrid-scale ACI effects between various subregions are related to convective conditions and model local errors. Additionally, compared to simulations with anthropogenic emissions turned off, subgrid-scale actual aerosol inhibits cumulative precipitation during a typical heavy rainfall event by 5.6%, aligning it with observations, associated with lower autoconversion at subgrid-scale and less available water vapor for grid-scale condensation, suggesting competitions between subgrid- and grid-scale cloud. This study demonstrates the importance of real-time subgrid-

scale ACI in NWP models and the necessity of multiscale ACI studies.

## 1 Introduction

Cloud plays an essential role in climate and weather by maintaining atmospheric radiation balance, regulating global precipitation, facilitating chemical reactions, etc. (Pruppacher and Klett, 1980; Seinfeld and Pandis, 2006; Fan et al., 2016). In the actual atmosphere, water vapor is hardly able to form cloud droplets spontaneously due to the free energy barrier until the

heterogeneous nucleation process is completed with the help of suspended aerosol particles (Seinfeld and Pandis, 2006; Sun



and Ariya, 2006). The perturbation of aerosol particles inevitably affects cloud properties, also known as aerosol-cloud interaction (ACI), including the Twomey effect (Twomey, 1977) and Albrecht effect (Albrecht, 1989). Due to the complexity of cloud and aerosol processes and their entangled nature, ACI is still subject to significant uncertainties in current climate projections and weather forecast (IPCC, 2021, 2013; Miltenberger et al., 2018; Baklanov et al., 2017). In the latest

Intergovernmental Panel on Climate Change (IPCC) report, ACI has the lowest confidence in effective radiative forcing estimates (IPCC, 2021).

Compared to the extensive research in the climate modeling community, ACI is less considered among various numerical weather prediction (NWP) models (Rosenfeld et al., 2014; Wang et al., 2014; Seinfeld et al., 2016). The NWP model runs daily in major regional operational centers worldwide and is primarily responsible for weather forecast. For a long time, operational

NWP models have been based on seven fundamental equations of atmospheric motion to predict future atmospheric states, with few considerations of the aerosol effect, especially ACI, on meteorology due to the cognitive and computing power (Grell and Baklanov, 2011; Sandu et al., 2013; Pleim et al., 2014; Baklanov et al., 2017). An aerosol climatology used in the NWP model may mitigate the forecast bias but cannot represent actual aerosol levels (Thompson and Eidhammer, 2014; Song and Zhang, 2011). The NWP models with "two-way" feedback between chemistry and meteorology (e.g., the Weather Research

and Forecasting model coupled with chemistry (WRF-Chem) and Weather Research and Forecasting and Community Multiscale Air Quality (WRF-CMAQ)) can fill this gap and have been wildly applied to multiscale studies to investigate the role of ACI in reducing radiation, cooling temperature, inhibiting or enhancing precipitation, etc. (Zhang et al., 2010; Grell and Baklanov, 2011; Wong, 2012; Makar et al., 2015; Zhang et al., 2015; Han et al., 2023). These studies have explicitly addressed that ACI has an essential influence on weather systems but have rarely focused on its feedback on NWP. With the

rapid development of supercomputing technology and the keen concerns about the impacts of anthropogenic activity on weather, the role of ACI in NWP is only beginning to be scrutinized in detail (Zhang et al., 2022; Zhang et al., 2024; Wang et al., 2021). For example, Zhang et al. (2024) show that coupling of real-time hygroscopic aerosol activation in the Thompson cloud microphysics scheme in an atmospheric chemistry model CMA_Meso5.1/CUACE improves the accuracy of predicted surface and vertical meteorological factors during the low-cloud period in winter of China.

To the best of our knowledge, almost all of the studies in this area focus on ACI at grid-scale. An important reason is that cloud microphysics schemes in NWP models include explicit cloud microphysics processes and aerosol activation, whereas cumulus convection schemes do not. Cumulus convection schemes in mesoscale NWP models are designed to characterize better subgrid-scale cloud processes that are not directly resolved (Arakawa, 2004; Plant, 2010), typically such as the Kain-Fritsch (KF) scheme (Kain and Fritsch, 1993) and the follow-up KFeta scheme (Kain, 2004), KFcup scheme (Berg et al., 2013) and

MSKF scheme (Zheng et al., 2016). These schemes are mass flux parameterizations that use grid-scale information to determine the conditions when convection occurs, include cloud models for both updrafts and downdrafts, and allow cumulus feedback for grid-scale cloud. Notably, during the periods of strong small-scale convections, only considering grid-scale ACI



potentially overlooks the effect of aerosol on convective clouds that are not resolvable at grid-scale, further affecting the assessment of the role of ACI in NWP. Cumulus convection schemes that include detailed cloud microphysical processes must be incorporated into the NWP model. To address aerosol-convective cloud simulations in global climate models (GCMs), Song and Zhang (2011) proposed a double-moment convective cloud microphysics scheme (SZ2011). Recently, Glotfelty et al. (2019) coupled the SZ2011 scheme with climatological aerosol concentration to the MSKF scheme in the WRF model and find that this system improves the simulation of cloud properties, which facilitates the study of ACI using the WRF model. It is worth noting that climatological aerosol that differs spatially and temporally from real-time predicted aerosol exacerbates uncertainty in ACI, especially at subgrid-scale, where the ACI appears to be more strongly represented at subgrid-scale compared to grid-scale (Glotfelty et al., 2019, 2020).

To investigate the impact of subgrid-scale ACI, a double-moment convective cloud microphysical scheme including real-time hygroscopic aerosol activation is coupled into the KFeta cumulus convection scheme in an atmospheric chemistry model CMA_Meso5.1/CUACE, the impact of the subgrid-scale ACI on the prediction of meteorological factors in summer in central and eastern China is investigated, and the role of anthropogenic aerosol activation at subgrid-scale in deep convective precipitation are further discussed.

## 2 Data

The data used in this paper are as follows: (1) Aerosol pollution observation data. Hourly $PM_{2.5}$ mass concentration ($\mu g\ m^{-3}$) comes from more than 1,300 air pollution stations of the Ministry of Ecology and Environment of the People's Republic of China. (2) Near-surface meteorological observation data. Hourly temperature at 2 m (T2m, ℃), relative humidity (RH) at 2 m (RH2m, %), wind speed at 10 m (WS10m, m s$^{-1}$), and 24 hours cumulative precipitation (PRE24h, mm) are provided by more than 5,000 automated weather stations of the China Meteorological Administration (CMA) (Figure 1). (3) Vertical meteorological observation data. Twice a day (00:00 and 12:00 UTC) temperature, RH, and WS are monitored by L-band radar from about 85 sounding stations of CMA (Figure 1). (4) Radiation observation data. Hourly surface downward shortwave radiation (SDSR, 0.01MJ m$^{-2}$) in the daytime is from more than 70 radiation stations of CMA (Figure 1). (5) Satellite data. Daily cloud fraction (CF, %), cloud liquid water path (CLWP, g m$^{-2}$), cloud optical thickness (COT), and aerosol optical depth (AOD) come from the Suomi National Polar-orbiting Partnership (SNPP) Visible Infrared Imaging Radiometer Suite (VIIRS). Daily SDSR (W m$^{-2}$) and surface downward longwave radiation (SDLR, W m$^{-2}$) come from the Clouds and the Earth's Radiant Energy System (CERES). The horizontal resolutions of these data are 1°×1°. Daily PRE24h (mm) from the Global Precipitation Measurement (GPM) program's Integrated Multi-satellitE Retrievals (IMERG) with a horizontal resolution of 10 km×10 km. (6) Re-analysis data. Final (FNL) operational global analysis and forecast data with a horizontal resolution of 0.25°×0.25° and a time interval of 6 hours come from the National Centers for Environmental Prediction (NECP)/National Center for



Atmospheric Research (NCAR). These data are primarily produced by the Global Data Assimilation System (GDAS), which

continuously collects observations from the Global Telecommunications System (GTS) and other sources. (7) Emission data.

The Multi-Resolution Emission Inventory for China (MEIC) anthropogenic emission data are provided by Tsinghua University,

including six sectors (power, industry, civil, transportation, and agriculture) and nine species ($SO_2$, $NO_x$, CO, NMVOC, $NH_3$,

$PM_{10}$, $PM_{2.5}$, BC, and OC).

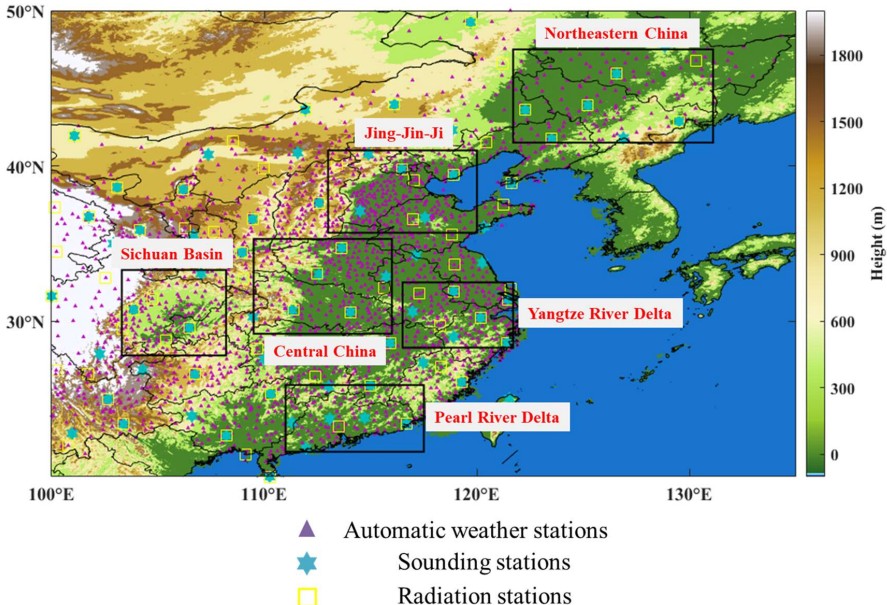

**Figure 1: The map and topographic height of the simulated domain. The purple triangles are the automatic weather stations, the**
**cyan hexagons are the sounding stations, the yellow boxes are the radiation sounding stations, and the black rectangles represent the**
**location of Northeastern China (NEC), Jing-Jin-Ji (JJJ), Sichuan Basin (SB), Central China (CC), Yangtze River Delta (YRD), and**
**Pearl River Delta (PRD), respectively.**

## 3 Model description and development

### 3.1 CMA_Meso5.1/CUACE model

The CMA_Meso/CUACE, independently developed by CMA, is online coupled with a mesoscale NWP model (China

Meteorological Administration Mesoscale model version 5.1 (CMA_Meso5.1)) with the atmospheric chemistry module

(Chinese Unified Atmospheric Chemistry Environment (CUACE)), which has been widely used for studying the ARI effects

on aerosol pollution, transboundary transport of air pollutants (Jiang et al., 2015), impacts of anthropogenic emissions on $PM_{2.5}$

changes (Wang et al., 2018; Zhang et al., 2020), visibility forecast (Peng et al., 2020; Han et al., 2024), fog-haze forecast (Zhou

et al., 2012; Wang et al., 2015b; Wang et al., 2015a; Li et al., 2023), etc. In this study, the latest quasi-operational version

CMA_Meso5.1/CUACE is used, and its specific updates can be found in the previous study (Wang et al., 2022).

The CMA_Meso5.1 is a continuous development of the GRAPES_Meso, mainly including Pre-processing and Quality Control,



Standard Initialization, Assimilating and Forecasting, and Post-processing, and is used to meet the operational needs of the short-term weather forecast in China (Chen and Shen, 2006; Chen et al., 2008; Zhang and Shen, 2008). In this model, the temporal, horizontal, and vertical discretization adopts the semi-implicit semi-Lagrangian scheme, Arakawa C-grid staggering, and Charney-Phillips staggering, respectively. This model also contains a series of physical parameterization schemes, such as radiation, boundary layer, near-surface layer, cumulus convection, and cloud microphysical schemes.

The CUACE is an atmospheric chemistry module that includes the emission treatment system, the gas and aerosol calculation processes, and the thermodynamic equilibrium module(Zhou et al., 2012; Wang et al., 2015b). There are seven types of aerosol: sulfates (SF), sand/dust (SD), black carbon (BC), organic carbon (OC), sea salts (SS), nitrates (NI), and ammonium (AM). All types of aerosol radii except AM are categorized into 12 bins ranging from 0.005-20.48 µm. Aerosol calculation processes include hygroscopic growth, wet and dry deposition, chemical transformations, coagulation, etc. The 63 species of gases in the CUACE are calculated and updated by 21 photochemical and 136 gas-phase chemical reactions.

### 3.2 Grid-scale ACI

Before dealing with subgrid-scale ACI, it is necessary to describe the grid-scale ACI implemented based on the double-moment Thompson cloud microphysics scheme in the current model. The original assumed cloud droplets number concentration (100 $cm^{-3}$) in the Thompson cloud microphysics scheme is replaced by the predicted value, which is determined based on the activation fraction of real-time calculated hygroscopic aerosol (OC, SS, SF, NT, and AM) in CUACE by the looking-up table; the fixed cloud water (10 µm) and cloud ice (80 µm) radius in the Goddard shortwave radiation scheme is replaced by diagnosed values in the Thompson cloud microphysics scheme. More detailed descriptions can be found in the previous study (Zhang et al., 2022). In this study, we do not make an extra consistent treatment of the grid-scale ACI because of the ability to understand the impact of subgrid-scale ACI and the convenience of comparison with the previous study.

### 3.3 Implementation of subgrid-scale ACI

### 3.3.1 Coupling of the double-moment microphysics parameterization scheme for convective cloud in the KFeta cumulus convection scheme

Optional cumulus convection parameterization schemes in the current model include the BMJ (Betts, 1986; Betts and Miller, 1986; Janjić, 1994), KFeta (Kain, 2004), NSAS (Han and Pan, 2011), and Tiedtke (Tiedtke, 1989) schemes. To implement subgrid-scale ACI, an efficient double-moment microphysics parameterization scheme for convective cloud is coupled into the commonly used KFeta cumulus convection scheme.

The KFeta scheme is a typical cumulus convection scheme used in the mesoscale NWP model, whose fundamental framework is derived initially from the Fritsch-Chappell convective parameterization scheme (Fritsch and Chappell, 1980). The classic KF scheme (Kain and Fritsch, 1993) has evolved through a series of modifications into the KFeta scheme, including imposed



minimum entrainment rate, variable cloud radius, variable minimum cloud-depth threshold, allowed shallow convection, etc.

(Kain, 2004). However, its treatment of convective cloud microphysical processes is rather crude, especially for the

transformations between the various hydrometeors within the convective cloud. At the same time, it is a mass-flux

parameterization scheme, which can correspond well to the double-moment microphysics parameterization scheme for

convective cloud.

This double-moment microphysics parameterization scheme for convective cloud is proposed by Song and Zhang (2011) to

improve the performance of convective cloud interacting with stratiform cloud and aerosol in GCMs. The mixing ratio and

number concentration of cloud water, cloud ice, rain, and snow can be simultaneously predicted. Figure 2 shows the

microphysical processes of these four hydrometeors in the double-moment microphysics parameterization scheme, mainly

including autoconversion, freezing, accretion, self-collection, detrainment, fallout, aerosol activation, ice nucleation, etc. The

detailed control equations and microphysical processes calculations for each hydrometeor can be found in the previous study.

The real-time activation of aerosol as CCN to cloud droplets is carried out through the ARG2000 scheme (Abdul-Razzak and

Ghan, 2000; Abdul-Razzak et al., 1998), as detailed in the section 3.3.2. The current scheme does not include real-time ice

nucleation because the dust is not available in the CUACE. The ice crystals number concentration can be derived using the

equation (1) proposed by Cooper (1986):

$$\text{sub\_Ni} = 0.005e^{0.304(273.15-\ )} \tag{1},$$

where sub_Ni is the ice crystals number concentration (/L) and T is the simulated ambient temperature (K) at subgrid-scale. It

should be noted that ice crystals can only form when the supersaturation with respect to ice exceeds 5%, or the supersaturation

with respect to water exceeds 0 and ambient temperature < -5 ℃, consistent with that in the Thompson cloud microphysics

scheme (Thompson and Eidhammer, 2014). Considering reducing the complexity of the code and additional errors, we directly

couple the SZ2011 scheme into the KFeta scheme via a one-to-one correspondence of specific values, such as cloud water

mixing ratio, cloud ice mixing ratio, rate of production of precipitation, and rate of production of snow.

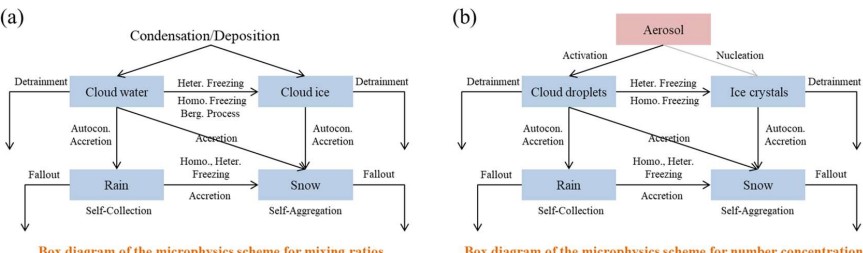


**Figure 2: Box diagram of microphysical processes for various hydrometeors mixing ratio (a) and number concentration (b) in the SZ2011 double-moment microphysics parameterization scheme for convective cloud. The real-time ice nucleation is not available.**

### 3.3.2 The real-time aerosol activation process

To implement real-time aerosol activation as CCN at subgrid-scale, the subgrid-scale cloud droplets number concentration



from the ARG2000 scheme (Abdul-Razzak and Ghan, 2000), driven by predicted hygroscopic aerosol in the CUACE, is integrated into the KFeta scheme with SZ2011 parameterization (Figure 3). The ARG2000 scheme is an activation scheme of aerosol with divided-component and divided-size, and is widely used in mesoscale NWP models. This parameterization is suitable for seven types of aerosol with 12 bins predicted by the CUACE module and described as following equations:

$$\text{sub\_Nc} = \sum_{num=1}^{49} N_{anum} \frac{1}{2} \left[ 1 - \text{erf} \left( \frac{2\ln(S_{mnum}/S_{max})}{3\sqrt{2}\ln\sigma_{num}} \right) \right] \tag{2},$$

$$S_{max} = \frac{1}{\left\{ \sum_{num=1}^{49} \frac{1}{S_{mnum}^2} \left[ (0.5e^{2.5\ln^2\sigma_{num}}) \left( \frac{\zeta}{\eta_{num}} \right)^{1.5} + (1+0.25\ln\sigma_{num}) \left( \frac{S_{mnum}^2}{\eta_{num}+3\zeta} \right)^{0.75} \right] \right\}^{0.5}} \tag{3},$$

$$S_{mnum} = \frac{2}{\sqrt{b_{num}}} \left( \frac{3.29 \times 10^{-7}}{3r_{num}T} \right)^{1.5} \tag{4},$$

where in the equation (2), sub_Nc is the subgrid-scale cloud droplets number concentration (kg$^{-1}$) generated by activation, $N_{anum}$ is the aerosol number concentration (kg$^{-1}$), $S_{max}$ is the maximum supersaturation, $S_{mnum}$ is the critical supersaturation for aerosol activation, $\sigma_{num}$ is the aerosol geometric standard deviation, erf is the Gaussian error function, and num is the aerosol type ranging from 1 to 49 (Table 1). $S_{max}$ can be solved by the equation (3), where $\zeta$ and $\eta$ are two dimensionless parameters given by Abdul-Razzak and Ghan (2000). $S_{mnum}$ can be solved by the equation (4), where $b_{num}$ is the aerosol hygroscopicity parameter, $r_{num}$ is the aerosol mean radius of (μm), and T is the ambient temperatur (K). In general, the solution of the activation fraction requires inputs of meteorological factors and aerosol parameters. Meteorological factors include atmospheric vertical velocity, temperature, etc., which can be provided in real-time by the CMA_Meso5.1 model. Aerosol parameters include aerosol number concentration, mass concentration, geometric standard deviation, density, and size. The CUACE module only outputs the aerosol mass mixing ratio, not the number concentration. Under the assumption that aerosol particles are spherical, each type of aerosol number concentration is obtained by the following equation (5):

$$N_{anum} = \text{tracer}_{num} / \left( \frac{4}{3} * \pi * r_{num}^3 * \rho_{num} \right) \tag{5},$$

where tracer$_{num}$ is the aerosol mass mixing ratio (kg kg$^{-1}$) generated by the CUACE and $\rho_{num}$ is the aerosol density (g cm$^{-3}$). All other aerosol parameters are preset: the density and radius are shown in Table 1; the geometric standard deviation is set to 2.0 for all types of aerosol; and the hygroscopicity parameters are set to 0.2, 1.28, 0.61, 0.67, and 0.64 for OC, SS, SF, NT, and AM, respectively. The hygroscopicity parameter for OC is slightly higher than the typical value of 0.1, which was attributed to the fact that the region of China is frequently hazed (Petters and Kreidenweis, 2007; Che et al., 2017). The hygroscopicity parameters of SS, SF, NT, and AM are similar to other studies (Kim et al., 2021; Morales Betancourt and Nenes, 2014; Petters and Kreidenweis, 2007). Identical to the grid-scale ACI mechanism, BC and SD, two non-hygroscopic aerosol, are not used as the subgrid-scale aerosol to be activated. It should be noted that cloud droplets can only form when the supersaturation with respect to water exceeds 0.

**Table 1: The specific values of the tracer number, aerosol types, mean radius (μm), density (g cm$^{-3}$), geometrical standard deviation (GSD), and hygroscopicity parameter.**



| Tracer number | Aerosol types | Radius | Density | GSD | Hygroscopicity |
|---|---|---|---|---|---|
| 1 | OC1 | 0.0075 | 1.30 | 2.0 | 0.2 |
| 2 | OC2 | 0.015 | 1.30 | 2.0 | 0.2 |
| 3 | OC3 | 0.03 | 1.30 | 2.0 | 0.2 |
| 4 | OC4 | 0.06 | 1.30 | 2.0 | 0.2 |
| 5 | OC5 | 0.12 | 1.30 | 2.0 | 0.2 |
| 6 | OC6 | 0.24 | 1.30 | 2.0 | 0.2 |
| 7 | OC7 | 0.48 | 1.30 | 2.0 | 0.2 |
| 8 | OC8 | 0.96 | 1.30 | 2.0 | 0.2 |
| 9 | OC9 | 1.92 | 1.30 | 2.0 | 0.2 |
| 10 | OC10 | 3.84 | 1.30 | 2.0 | 0.2 |
| 11 | OC11 | 7.68 | 1.30 | 2.0 | 0.2 |
| 12 | OC12 | 15.36 | 1.30 | 2.0 | 0.2 |
| 13 | SS1 | 0.0075 | 2.17 | 2.0 | 1.28 |
| 14 | SS2 | 0.015 | 2.17 | 2.0 | 1.28 |
| 15 | SS3 | 0.03 | 2.17 | 2.0 | 1.28 |
| 16 | SS4 | 0.06 | 2.17 | 2.0 | 1.28 |
| 17 | SS5 | 0.12 | 2.17 | 2.0 | 1.28 |
| 18 | SS6 | 0.24 | 2.17 | 2.0 | 1.28 |
| 19 | SS7 | 0.48 | 2.17 | 2.0 | 1.28 |
| 20 | SS8 | 0.96 | 2.17 | 2.0 | 1.28 |
| 21 | SS9 | 1.92 | 2.17 | 2.0 | 1.28 |
| 22 | SS10 | 3.84 | 2.17 | 2.0 | 1.28 |
| 23 | SS11 | 7.68 | 2.17 | 2.0 | 1.28 |
| 24 | SS12 | 15.36 | 2.17 | 2.0 | 1.28 |
| 25 | SF1 | 0.0075 | 1.79 | 2.0 | 0.61 |
| 26 | SF2 | 0.015 | 1.79 | 2.0 | 0.61 |
| 27 | SF3 | 0.03 | 1.79 | 2.0 | 0.61 |
| 28 | SF4 | 0.06 | 1.79 | 2.0 | 0.61 |
| 29 | SF5 | 0.12 | 1.79 | 2.0 | 0.61 |
| 30 | SF6 | 0.24 | 1.79 | 2.0 | 0.61 |
| 31 | SF7 | 0.48 | 1.79 | 2.0 | 0.61 |
| 32 | SF8 | 0.96 | 1.79 | 2.0 | 0.61 |
| 33 | SF9 | 1.92 | 1.79 | 2.0 | 0.61 |
| 34 | SF10 | 3.84 | 1.79 | 2.0 | 0.61 |
| 35 | SF11 | 7.68 | 1.79 | 2.0 | 0.61 |
| 36 | SF12 | 15.36 | 1.79 | 2.0 | 0.61 |
| 37 | NT1 | 0.0075 | 1.77 | 2.0 | 0.67 |
| 38 | NT2 | 0.015 | 1.77 | 2.0 | 0.67 |
| 39 | NT3 | 0.03 | 1.77 | 2.0 | 0.67 |
| 40 | NT4 | 0.06 | 1.77 | 2.0 | 0.67 |
| 41 | NT5 | 0.12 | 1.77 | 2.0 | 0.67 |
| 42 | NT6 | 0.24 | 1.77 | 2.0 | 0.67 |
| 43 | NT7 | 0.48 | 1.77 | 2.0 | 0.67 |
| 44 | NT8 | 0.96 | 1.77 | 2.0 | 0.67 |
| 45 | NT9 | 1.92 | 1.77 | 2.0 | 0.67 |
| 46 | NT10 | 3.84 | 1.77 | 2.0 | 0.67 |
| 47 | NT11 | 7.68 | 1.77 | 2.0 | 0.67 |
| 48 | NT12 | 15.36 | 1.77 | 2.0 | 0.67 |
| 49 | AM | 0.06 | 1.69 | 2.0 | 0.64 |


### 3.3.3 The feedback of subgrid-scale cloud to radiation

In order to represent the impact of subgrid-scale ACI on radiation, this study completed the feedback of subgrid-scale cloud



properties on radiation: coupling subgrid-scale CF, cloud water mixing ratio (Qc), ice mixing ratio (Qi), cloud wate effective

radius (Rc), and cloud ice effective radius (Ri) into the Goddard shortwave radiation scheme (Figure 3). It should be noted that

the grid-scale CF, Qc, and Qi are the default inputs to the Goddard shortwave radiation scheme, and Rc and Ri at grid-scale

based on the diagnostics of the Thompson cloud microphysics scheme have also been coupled into the radiation scheme in the

previous study (Zhang et al., 2022). The subgrid-scale CF is calculated with reference to CAM5 (Neale et al., 2010; Xu and

Krueger, 1991), where the CF for deep convection and shallow convection have been estimated separately in the KFeta scheme.

These two types of CF have been added directly to the grid-scale CF with keeping the total CF range between 0 and 1. The

subgrid-scale Qc and Qi are derived from the SZ2011 scheme and are combined with the grid-scale Qc and Qi with reference

to the previous study(Alapaty et al., 2012). The subgrid-scale Rc and Ri are also derived from the SZ2011 scheme, which is

combined with the grid-scale Rc and Ri based on the study of Thompson et al. (2016) and Glotfelty et al. (2019) (Thompson

et al., 2016; Glotfelty et al., 2019). The adjusted CF, Qc, Qi, Rc, and Ri in the Goddard shortwave radiation scheme

simultaneously incorporate cloud properties at both grid-scale and subgrid-scale.

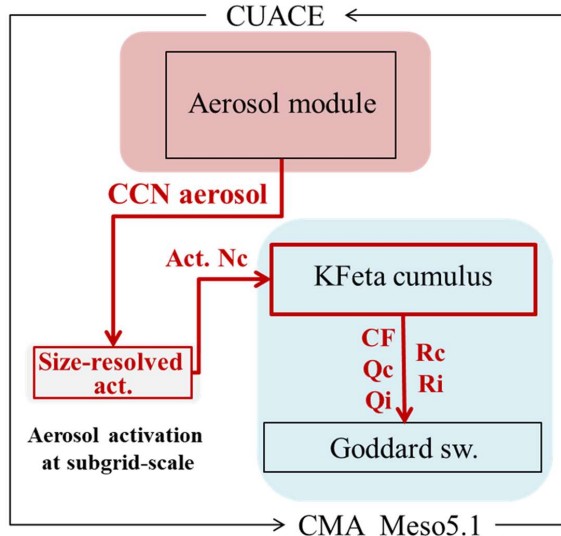

**Figure 3: The diagram of subgrid-scale aerosol–cloud-radiation interaction in the CMA_Meso5.1/CUACE model.**

## 4 Model configurations and experimental design

In this study, two sets of experiments are conducted using the CMA_Meso5.1/CUACE model to investigate the impact of

subgrid-scale ACI. In the first set of experiments, the NO-ACI$_{sub}$ and ACI$_{sub}$ experiments are included to focus on the summer

of 2016 (June represents the summer season), when convection occurs more frequently in China, and the water vapor

conditions are better, focusing on the NEC, JJJ, SC, CC, YRD, and PRD regions (Figure 1). The average of these six regions

is used to represent the whole central and eastern China. The NO-ACI$_{sub}$ experiment is the control experiment, and the model



configurations are shown in Table 2. These settings are the same as the previous study (Zhang et al., 2022). The ACI$_{sub}$

experiment contains all the treatments of the relevant subgrid-scale ACI mechanisms in the section 3.3, except that the other

settings are the same as the NO-ACI$_{sub}$ experiment (Table 3). The difference between the ACI$_{sub}$ and NO-ACI$_{sub}$ experiments

shows the impact of subgrid-scale ACI on the performance of predicted meteorological factors in the current model. The

simulated periods of both experiments are from 29 May to 30 June 2016, with a forecast time of 24 hours, a time step of 100

s, and an output interval of 1 hour. The 72 hours pre-simulations are used to keep a balance between the chemical initial field

and the meteorological field, which are treated as the spin-up time. In the second set of experiments, the ACI$_{sub}$-DC and

CACI$_{sub}$-DC experiments are included to study the impact of anthropogenic aerosol on cloud and precipitation via the subgrid-

scale ACI mechanism, mainly for a typical deep convective heavy precipitation process (from 26 to 29 June 2016). The settings

of the ACI$_{sub}$-DC experiment are the same as those of the ACI$_{sub}$ experiments except for the fixed cloud droplets number

concentration (300 cm$^{-3}$) in the Thompson cloud microphysics scheme, which can prevent the additional uncertainties from

anthropogenic aerosol affecting the grid-scale ACI. In the CACI$_{sub}$-DC experiment, the MEIC anthropogenic emissions are

turned off in the model, and other settings are the same as those of the ACI$_{sub}$-DC experiment (Table 3). The difference between

ACI$_{sub}$-DC and CACI$_{sub}$-DC indicates the impact of anthropogenic aerosol via the subgrid-scale ACI. The simulated periods

of both experiments are from 23 to 30 June 2016, with a forecast time of 48 hours. The first 72 hours of simulations are also

treated as the spin-up time. The initial field and boundary conditions for meteorology are provided by the FNL data, which are

same as the time period simulated for each set of experiments. The anthropogenic emission data in June 2016 entered into the

model are from MEIC.

**Table 2: Model configurations.**

| Parameters and schemes | Setting |
|---|---|
| Simulated domain | 100°-135°E, 20°-50°N |
| Horizontal resolution | 10 km |
| Vertical stratification | 49 levels (from ground to 31 km) |
| Cumulus convective scheme | KFeta (Kain, 2004) |
| Land surface scheme | Noah (Ek et al., 2003) |
| Short-wave radiation scheme | Goddard (Chou et al., 1998) |
| Long-wave radiation scheme | RRTM (Mlawer et al., 1997) |
| Cloud microphysics scheme | Thompson (Thompson et al., 2008) |
| Gas-phase chemistry scheme | RADM2 (Stockwell et al., 1990) |
| Boundary layer scheme | MRF (Hong & Pan, 1996) |
| Near-surface scheme | SFCLAY (Pleim, 2006) |
| Aerosol scheme | CUACE (Gong & Zhang, 2008) |

**Table 3: Descriptions of multiple sensitivity experiments.**

| Experiment | Description |
|---|---|
| NO-ACI$_{sub}$ | Controlled experiment without subgrid-scale ACI |
| ACI$_{sub}$ | Same as NO-ACI$_{sub}$, but including subgrid-scale ACI |



| | |
|---|---|
| $ACI_{sub}$-DC | Same as $ACI_{sub}$, but for a deep convective process and fixing the cloud droplets number concentration in the Thompson cloud microphysics scheme as 300 cm$^{-3}$ |
| $CACI_{sub}$-DC | Same as $ACI_{sub}$-DC, but turning off MEIC anthropogenic emissions |


## 5. Results and discussions

### 5.1 Evaluations of $PM_{2.5}$ mass concentration and AOD

To assess the performance of the CMA_Meso5.1/CUACE model in aerosol prediction, Figure 4 shows the comparisons of spatial distributions of the observed and simulated time average $PM_{2.5}$ mass concentration and AOD in June 2016. As shown,

the observed $PM_{2.5}$ mass concentration over widespread areas of the domain is almost below 75 μg m$^{-3}$, with the regional average $PM_{2.5}$ mass concentration of 26.4, 47.9, 33.6, 37.3, 35.8, and 19.4 μg m$^{-3}$ in the NEC, JJJ, SB, CC, YRD, and PRD. The model reproduces the spatial distribution of the high-value and low-value areas of $PM_{2.5}$ mass concentration and captures the magnitude of $PM_{2.5}$ mass concentration at most air quality monitoring stations. The mean bias (MB) of regional average $PM_{2.5}$ mass concentration is -12.2, -16.3, 3.2, 0.9, -2.9, and -2.9 μg m$^{-3}$ in the NEC, JJJ, SB, CC, YRD, and PRD, respectively.

The significantly underestimated $PM_{2.5}$ mass concentration in the NEC and JJJ region is possibly related to underestimated anthropogenic emissions, inadequate representation of aerosol chemical reaction processes, etc. The AOD represents the column-integrated aerosol properties here. The VIIRS data show that the regional average AOD is 0.44, 0.53, 0.16, 0.38, 0.44, and 0.17 in the NEC, JJJ, SB, CC, YRD, and PRD, respectively. The model seems to capture some high-value and low-value areas of AOD well in the south of the domain (e.g., the regional average bias is -0.05, -0.04, and 0.02 in the CC, YRD, and

PRD) but significantly underestimates AOD in the north of the domain (e.g., the regional average MB is -0.31 and -0.27 in the NEC and JJJ). This substantially underestimated AOD in the NEC and JJJ region is mainly caused by the underestimated $PM_{2.5}$ mass concentration. It is noted that in the SC with MB of 0.15 for AOD, the overestimations are possibly related to the errors of the VIIRS data in complex terrain (Ali et al., 2019; Kainan et al., 2019). Compared with other studies or models, the CMA_Meso5.1/CUACE model has a similar performance in predicting $PM_{2.5}$ mass concentration and AOD over China in

summer (Werner et al., 2019; Wang et al., 2021; He et al., 2022). This study's relatively reliable aerosol simulation performance can ensure the scientificity of further subgrid-scale ACI studies.



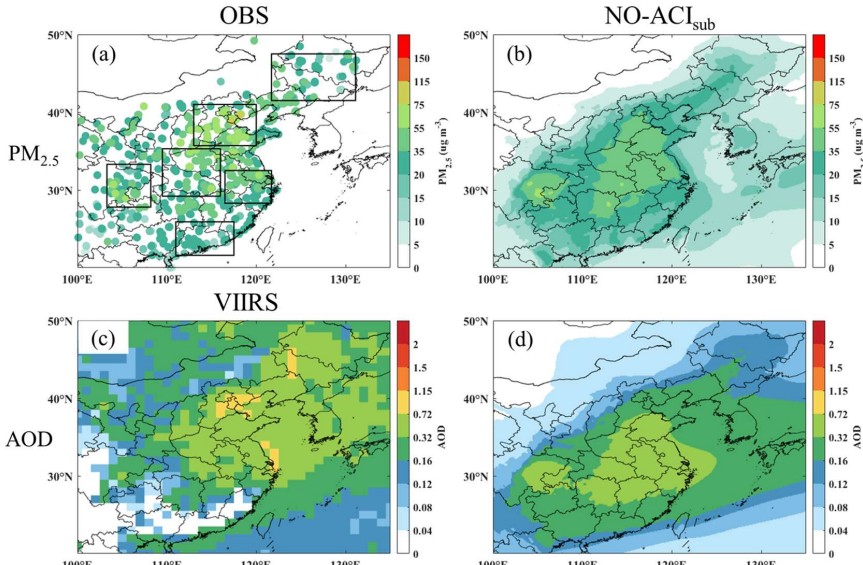

**Figure 4: Spatial distribution of time average PM$_{2.5}$ (a and b) and AOD (c and d) in June 2016 from the NO-ACI$_{sub}$ experiment compared against the observations and VIIRS estimates.**

**5.2 The impact of subgrid-scale ACI on the prediction of meteorological factors**

**5.2.1 Cloud properties**

Figure 5 compares the time average cloud properties in June 2016 between simulations and the VIIRS data. From the VIIRS data, CF, CLWP, and COT all show a distribution of high in the south and low in the north in June 2016 in the central and eastern China, which is mainly related to the higher RH in the south. Both the NO-ACI$_{sub}$ and ACI$_{sub}$ experiments reproduce

the spatial distribution of cloud properties, but the simulated CF, CLWP, and COT all have some bias in magnitude, and the specific statistics (MB, mean absolute error (MAE), root mean square error (RMSE) and correlation coefficient (R)) can be seen in Table 4. For CF, the model performs better in the north but shows a significant overestimation in the south (e.g., the MB of CF in the PRD for the NO-ACI$_{sub}$ and ACI$_{sub}$ experiments reach 0.17 and 0.16, respectively), which is mainly related to the overestimation of high CF in the south (figure omitted). The NO-ACI$_{sub}$ experiment also significantly underestimated

the CLWP (COT) over the whole domain, where the MB in the NEC, JJJ, SC, CC, YRD, and PRD are -138.7 (-15), -131.2 (-18.2), -148.4 (-10.2), -159.2 (-12.3), -174.3 (-10.2), and -105.3 (-6.6) g m$^{-2}$. Compared to the NO-ACI$_{sub}$ experiment, the subgrid-scale ACI significantly increases CLWP, 12.1 especially in the southern regions of China (e.g., the YRD), where convection occurs more frequently, and water vapor conditions are better. In addition, the coverage of cloud water in the model coupled with the subgrid-scale ACI is larger and contains some areas that are not saturated with respect to water at grid-scale.

Correspondingly, the MB of CLWP (COT) in the NEC, JJJ, SC, CC, YRD, and PRD for the ACI$_{sub}$ experiment are -58.8 (-3), -89.3 (-10.5), -50.2 (3.6), -82.7 (0.2), -56.3 (9.1), and 47.4 (14) g m$^{-2}$, respectively. It can be seen that the subgrid-scale ACI




generally improves the underestimated CLWP in these six regions (especially in the YRD), resulting in a 55.1% (from 142.9 to 64.1 g m$^{-2}$) decrease in the overall MB averaged over the six regions, which is closer to the VIIRS data. Slightly different from CLWP, subgrid-scale ACI does not generally make a decrease in the MB of COT in each region (e.g., the absolute MB of COT in the PRD increases by 7.4), which suggests that the impact of subgrid-scale ACI on the accuracy of NWP also depends on the local errors of model itself. Even if the subgrid-scale ACI mechanism is considered in the model, the simulations of cloud properties still have some bias. The problem of poorly simulated cloud properties is relatively common in both global and regional NWP models (Lauer and Hamilton, 2013; Wang et al., 2021; Glotfelty et al., 2019), which is one of the key issues that need to be urgently solved in the current scientific community. Overall, the ACI$_{sub}$ experiment shows relatively better performance compared to the NO-ACI$_{sub}$ experiment in June 2016 in the central and eastern China for cloud properties.

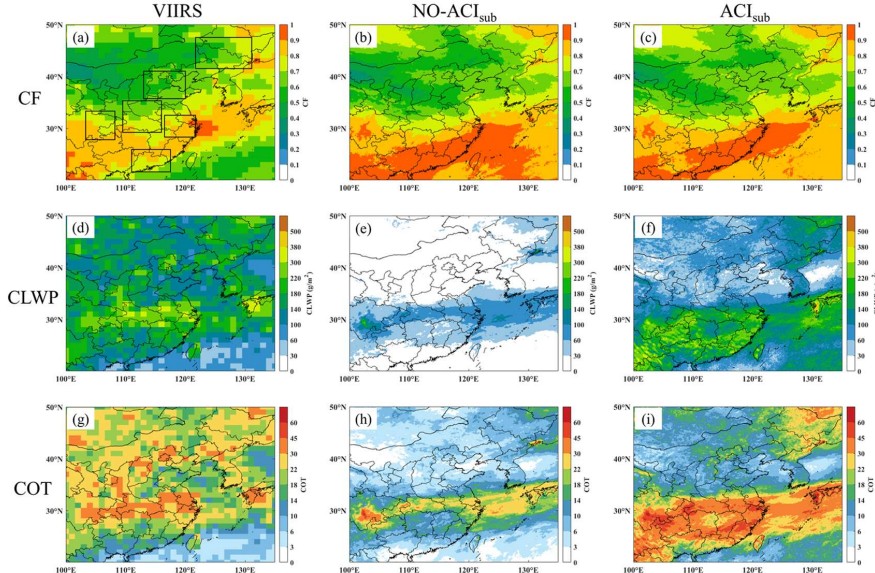

**Figure 5: The spatial distribution of time average (a-c) CF, (d-f) CLWP, and (g-i) COT in June 2016. The left, middle, and right column is the VIIRS, NO-ACI$_{sub}$, and ACI$_{sub}$ experiment, respectively.**

**Table 4: Statistics of simulated CF, CLWP (g m$^{-2}$), COT, SDSR (W m$^{-2}$), and SDLR (W m$^{-2}$) by the NO-ACI$_{sub}$ and ACI$_{sub}$ experiment.**

| Variable | Area | Satellites | NO-ACI$_{sub}$ | | | | | ACI$_{sub}$ | | | | |
|---|---|---|---|---|---|---|---|---|---|---|---|---|
| | | Mean Obs | Mean Sim | MB | MAE | RMSE | R | Mean Sim | MB | MAE | RMSE | R |
| CF | NEC | 0.67 | 0.65 | -0.02 | 0.09 | 0.12 | 0.83 | 0.68 | 0.01 | 0.08 | 0.11 | 0.85 |
| | JJJ | 0.62 | 0.6 | -0.02 | 0.13 | 0.16 | 0.59 | 0.64 | 0.02 | 0.12 | 0.15 | 0.66 |
| | SB | 0.75 | 0.78 | 0.03 | 0.09 | 0.1 | 0.92 | 0.8 | 0.05 | 0.08 | 0.1 | 0.94 |
| | CC | 0.69 | 0.67 | -0.02 | 0.1 | 0.13 | 0.86 | 0.7 | 0.01 | 0.09 | 0.11 | 0.89 |
| | YRD | 0.82 | 0.84 | 0.02 | 0.08 | 0.12 | 0.77 | 0.84 | 0.02 | 0.07 | 0.11 | 0.8 |
| | PRD | 0.77 | 0.94 | 0.17 | 0.19 | 0.29 | 0.34 | 0.93 | 0.16 | 0.17 | 0.27 | 0.48 |
| CLWP | NEC | 164.1 | 25.4 | -138.7 | 115.8 | 126 | 0.76 | 105.3 | -58.8 | 41.6 | 51.5 | 0.82 |



| | | | | | | | | | | | | |
|---|---|---|---|---|---|---|---|---|---|---|---|---|
| | JJJ | 144.1 | 12.9 | -131.2 | 92.4 | 102.8 | 0.79 | 54.8 | -89.3 | 50 | 61.5 | 0.8 |
| | SB | 208.9 | 60.5 | -148.4 | 114.8 | 127.3 | 0.71 | 158.7 | -50.2 | 67.2 | 85.2 | 0.67 |
| | CC | 205.3 | 46.1 | -159.2 | 119.3 | 131.5 | 0.6 | 122.6 | -82.7 | 83.6 | 102.6 | 0.58 |
| | YRD | 241.9 | 67.6 | -174.3 | 153.4 | 180 | 0.7 | 185.6 | -56.3 | 86.2 | 115.6 | 0.71 |
| | PRD | 126.9 | 21.6 | -105.3 | 122.7 | 131 | 0.72 | 174.3 | 47.4 | 58.4 | 81.8 | 0.73 |
| COT | NEC | 23.2 | 8.2 | -15 | 13.1 | 14.5 | 0.67 | 20.2 | -3 | 6.5 | 8.2 | 0.71 |
| | JJJ | 22.8 | 4.6 | -18.2 | 12 | 13.5 | 0.70 | 12.3 | -10.5 | 7.2 | 8.6 | 0.72 |
| | SB | 28.3 | 18.1 | -10.2 | 12.5 | 14.8 | 0.69 | 31.9 | 3.6 | 16.4 | 20.9 | 0.67 |
| | CC | 26.4 | 14.1 | -12.3 | 12.4 | 14.4 | 0.79 | 26.2 | 0.2 | 14.7 | 19.8 | 0.80 |
| | YRD | 30.6 | 20.4 | -10.2 | 11.5 | 15.5 | 0.72 | 39.7 | 9.1 | 18.6 | 26.1 | 0.67 |
| | PRD | 13.4 | 6.8 | -6.6 | 8.8 | 9.5 | 0.78 | 27.4 | 14 | 14.3 | 21.1 | 0.75 |
| SDSR | NEC | 221.7 | 293.1 | 71.4 | 66.9 | 74.6 | 0.85 | 272.7 | 51 | 46.9 | 53.2 | 0.89 |
| | JJJ | 233.7 | 310.1 | 76.4 | 73.6 | 80.4 | 0.86 | 299.9 | 66.3 | 63.4 | 68.8 | 0.93 |
| | SB | 200.5 | 287.1 | 86.6 | 85.6 | 93.4 | 0.86 | 269.6 | 69.1 | 68.3 | 75.2 | 0.89 |
| | CC | 201.9 | 282.9 | 81 | 80.1 | 88.2 | 0.79 | 268 | 66.1 | 65.3 | 72.6 | 0.85 |
| | YRD | 165.4 | 265.9 | 100.5 | 98.9 | 103.7 | 0.87 | 242.1 | 76.7 | 75.1 | 78.5 | 0.93 |
| | PRD | 212.3 | 277 | 64.7 | 65.1 | 76.3 | 0.9 | 253 | 40.6 | 42.4 | 52.9 | 0.91 |
| SDLR | NEC | 359.4 | 353.4 | -6 | 7.3 | 8.8 | 0.96 | 358.5 | -0.9 | 5.4 | 6.6 | 0.97 |
| | JJJ | 375.4 | 369.1 | -6.3 | 8.1 | 9.8 | 0.95 | 373 | -2.4 | 6.1 | 7.3 | 0.96 |
| | SB | 388.1 | 393.9 | 5.8 | 6.7 | 8.1 | 0.95 | 396.4 | 8.3 | 8.4 | 9.7 | 0.96 |
| | CC | 399.4 | 398.5 | -0.9 | 5.6 | 7.3 | 0.95 | 400.7 | 1.3 | 4.6 | 5.9 | 0.97 |
| | YRD | 413.3 | 415.3 | 2 | 6.9 | 8.3 | 0.97 | 417.2 | 3.9 | 6.6 | 7.9 | 0.98 |
| | PRD | 424.8 | 427.6 | 2.8 | 3.8 | 4.7 | 0.93 | 431.1 | 6.3 | 6.4 | 7.3 | 0.93 |

### 5.2.2 Radiation Properties

Figure 6 compares the time average radiation properties in June 2016 between the simulations and CERES data. Influenced by cloud characteristics, the SDSR in June 2016 shows a low south and high north distribution, while the opposite is true for the SDLR. The NO-ACI$_{sub}$ and ACI$_{sub}$ experiments can reproduce the spatial distribution of the radiative properties. For SDLR,

this model has a good prediction performance. This is supported by relevant statistical indicators (Table 4). Compared to the NO-ACI$_{sub}$ experiment, the ACI$_{sub}$ experiment improves the underestimation of SDLR in the northern part of the domain (e.g., the MB of SDLR decreases from -6 and -6.3 W m$^{-2}$ to -0.9 and -2.4 W m$^{-2}$ in the NEC and JJJ, respectively), but further overestimates the SDLR in most of the southern regions (e.g., the MB increases from 2 and 2.8 W m$^{-2}$ to 3.9 and 6.3 W m$^{-2}$ in the YRD and PRD, respectively). For SDSR, there are significant overestimations for both the NO-ACI$_{sub}$ and ACI$_{sub}$

experiments (e.g., the MB reach up to 100.5 and 76.7 W m$^{-2}$ in the YRD), which may be related to the poor simulation performance of cloud properties by the commonly reported mesoscale NWP models (Lauer and Hamilton, 2013; Wang et al., 2021). Compared with the two experiments, the ACI$_{sub}$ experiment improved the overestimation of SDSR in the NO-ACI$_{sub}$ experiment to a certain extent, especially in the regions where CLWP and COT increase significantly (e.g., the YRD and PRD). Correspondingly, the MB of simulated SDSR averaged over the six regions decreases by 23.1% (from 80.1 to 61.6 W m$^{-2}$). To

more accurately compare the prediction performance of NO-ACI$_{sub}$ and ACI$_{sub}$ for SDSR, this study conducted a comparative assessment using hour-by-hour SDSR observations (Figure 7). Similar to the results of the two experiments compared with





the CERES data, the daytime SDSR simulated by the ACI$_{sub}$ experiment is closer to the observations than that by the NO-ACI$_{sub}$ experiment in general, with the MB in the NEC, JJJ, SC, CC, YRD, and PRD decreasing by 30.5, 16.1, 29.6, 23.2, 40.5, and 41.2 W m$^{-2}$. The decrease in the upper quartile of SDSR bias is larger than that in the lower quartile in all six typical

regions. The larger SDSR bias tends to appear in the midday to mid-afternoon period, which indicates that the improvement in the SDSR bias induced by the subgrid-scale ACI is mainly manifested in the midday to mid-afternoon period.

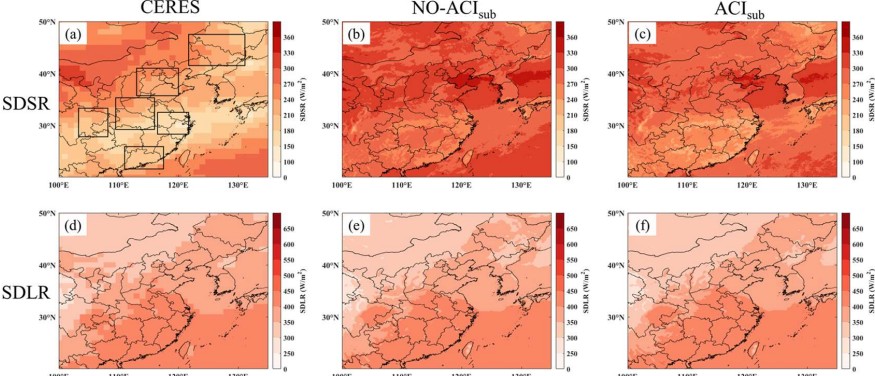

**Figure 6: The spatial distribution of time average (a-c) SDSR, (d-f) SDLR in June 2016. The left, middle, and right column is the CERES, NO-ACI$_{sub}$, and ACI$_{sub}$ experiment, respectively.**

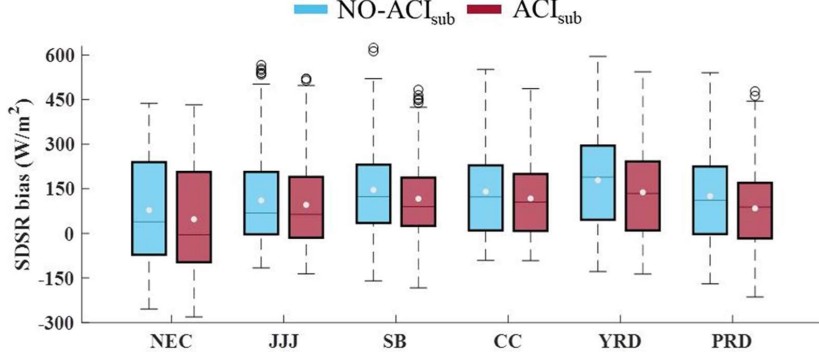


**Figure 7: Regional average bias of simulated daytime SDSR in NEC, JJJ, SB, CC, YRD, and PRD during the study period. The interquartile range is shown by boxes and with whiskers for the most extreme data points excluding outliers. The central lines and white dots present the median and mean values, respectively. The blue and red boxes are the values from the NO-ACI$_{sub}$ and ACI$_{sub}$ experiment, respectively.**

**5.2.3 Temperature**

Figure 8 shows the comparisons of the observed and simulated temperatures. For T2m, this model has a better performance overall, and the related statistical indicators (Table 5) also show that the model's simulation performance is in the middle compared with other studies or models (Bozzo et al., 2020; Wang et al., 2021; Gao et al., 2022). Compared with observations, both the NO-ACI$_{sub}$ and ACI$_{sub}$ experiments significantly overestimate T2m in most plains and underestimate T2m in some

mountainous areas, thus overestimating terrestrial T2m in the domain as a whole. Unlike other mesoscale NWP models that



usually exhibit overall negative regional MB of T2m in summer, the overall positive MB in the CMA_Meso5.1/CUACE model

may be related to the underestimated aerosol concentration, the selection of boundary layer schemes, etc. (Xie et al., 2012).

The T2m in the ACI$_{sub}$ experiment is smaller than that in the NO-ACI$_{sub}$ experiment due to the increase in COT and decrease

in SDSR caused by the subgrid-scale ACI, which correspondingly reduces the positive MB of T2m in the vast majority of

regions, with the MB of T2m averaged over the six regions decreasing by 40% (from 0.75°C to 0.4°C). Other statistical

indicators also show the improved performance of T2m simulations in the ACI$_{sub}$ experiment (Table 5). However, for the SB

region with large negative MB of T2, the cooling effect of subgrid-scale ACI further leads to an increase in the negative MB

(from -0.2°C to -0.7°C), but the T2m correlation coefficients have increased in this region. Also, this model reproduces the

vertical profile of temperature better, but the six typical regions generally have significant positive MB below about 900 hPa

(Figure 8(f)). Temperature over most of the air layers simulated by the ACI$_{sub}$ experiment is closer to observations than that by

the NO-ACI$_{sub}$ experiment, with the ranges of mean absolute error skill score (MAESS) of temperatures from 2 m to 500 hPa

in the NEC, JJJ, SC, CC, YRD, and PRD being -2% to 17%, 5% to 0.22%, 3% to 25%, -8% to 22%, 1% to 33%, and 5% to

32% (Figure 11).

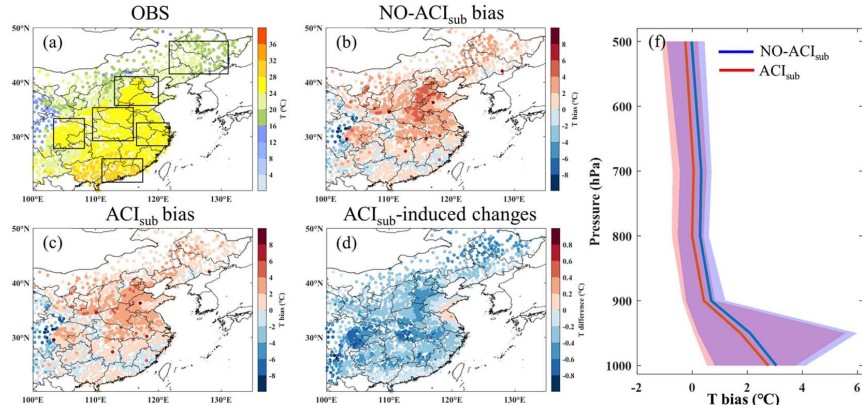

**Figure 8: The spatial distribution of time average T2m and the vertical profiles of MB of temperature in June 2016. (a) The observations. (b) The MB of T2m in the NO-ACI$_{sub}$ experiment. (c) The MB of T2m in the ACI$_{sub}$ experiment. (c) The difference of T2m between the ACI$_{sub}$ and NO-ACI$_{sub}$ experiment. (f) The vertical profiles of MB of temperature in the NO-ACI$_{sub}$ and ACI$_{sub}$ experiment. In the (f), the shadings are the spread of MB of temperature in six regions, and the solid lines are their average results.**

**Table 5: Statistics of simulated T2m (°C), RH2m (%), WS10m (m s⁻¹), and 24 hours cumulative precipitation (PRE24h, mm) by the**
**NO-ACI$_{sub}$ and ACI$_{sub}$ experiment.**

| Variable | Area | Satellites | NO-ACI$_{sub}$ | | | | | ACI$_{sub}$ | | | | |
|---|---|---|---|---|---|---|---|---|---|---|---|---|
| | | Mean obs | Mean sim | MB | MAE | RMSE | R | Mean sim | MB | MAE | RMSE | R |
| T2m | NEC | 19.5 | 20.5 | 1 | 1.7 | 2.3 | 0.84 | 20.2 | 0.7 | 1.4 | 2 | 0.87 |
| | JJJ | 24.2 | 25.1 | 0.9 | 1.8 | 2.1 | 0.9 | 24.8 | 0.6 | 1.5 | 1.8 | 0.93 |
| | SB | 25.4 | 25.2 | -0.2 | 1.3 | 1.8 | 0.86 | 24.7 | -0.7 | 1.4 | 1.8 | 0.88 |
| | CC | 25.3 | 26.7 | 1.4 | 1.7 | 2.2 | 0.91 | 26.4 | 1.1 | 1.3 | 1.9 | 0.93 |





|  |  |  |  |  |  |  |  |  |  |  |  |
|---|---|---|---|---|---|---|---|---|---|---|---|
|  | YRD | 24.6 | 25.7 | 1.1 | 1.6 | 1.9 | 0.9 | 25.4 | 0.8 | 1.3 | 1.6 | 0.92 |
|  | PRD | 27.8 | 27.9 | 0.1 | 1.4 | 1.7 | 0.77 | 27.7 | -0.1 | 1.3 | 1.6 | 0.81 |
| RH2m | NEC | 68.5 | 52.1 | -16.4 | 16.9 | 18 | 0.9 | 55.2 | -13.3 | 13.6 | 15 | 0.91 |
|  | JJJ | 60 | 44.6 | -15.4 | 17 | 17.1 | 0.92 | 47.1 | -12.8 | 14.4 | 14.5 | 0.92 |
|  | SB | 73.6 | 58.8 | -14.8 | 14.2 | 16.7 | 0.85 | 62.1 | -11.5 | 10.8 | 13.5 | 0.86 |
|  | CC | 72.1 | 55 | -17.1 | 16.6 | 18.5 | 0.86 | 57.3 | -14.8 | 14.1 | 16.4 | 0.87 |
|  | YRD | 84.2 | 72 | -12.2 | 12.2 | 13.8 | 0.81 | 74.2 | -10 | 9.9 | 11.4 | 0.86 |
|  | PRD | 83.4 | 76.7 | -6.7 | 7.6 | 9.5 | 0.79 | 78.1 | -5.3 | 6.5 | 8 | 0.84 |
| WS10m | NEC | 2.5 | 3.2 | 0.7 | 1 | 1.2 | 0.38 | 3.1 | 0.6 | 0.9 | 1.1 | 0.4 |
|  | JJJ | 2.2 | 4 | 1.8 | 1.8 | 2.1 | 0.5 | 3.9 | 1.7 | 1.7 | 2 | 0.51 |
|  | SB | 1.6 | 2.9 | 1.3 | 1.4 | 1.7 | 0.3 | 3 | 1.4 | 1.5 | 1.8 | 0.33 |
|  | CC | 2 | 3 | 1 | 1.1 | 1.4 | 0.47 | 3.1 | 1.1 | 1.2 | 1.5 | 0.5 |
|  | YRD | 1.9 | 3.5 | 1.6 | 1.7 | 2 | 0.2 | 3.5 | 1.6 | 1.6 | 1.9 | 0.22 |
|  | PRD | 1.9 | 4.1 | 2.2 | 2.2 | 2.6 | 0.26 | 3.9 | 2.0 | 2.0 | 2.4 | 0.28 |
| PRE24h | NEC | 4.6 | 2.6 | -2 | 2 | 2.9 | 0.93 | 2.9 | -1.7 | 1.8 | 2.4 | 0.94 |
|  | JJJ | 3.5 | 1.7 | -1.8 | 1.8 | 3.2 | 0.88 | 2.0 | -1.5 | 1.6 | 2.5 | 0.92 |
|  | SB | 6.2 | 4.7 | -1.5 | 3.2 | 4.5 | 0.73 | 7.6 | 1.4 | 3.1 | 4.8 | 0.78 |
|  | CC | 6.4 | 3.3 | -3.1 | 3.3 | 5.2 | 0.87 | 4.9 | -1.5 | 2.5 | 4 | 0.89 |
|  | YRD | 11 | 6.2 | -4.8 | 5.5 | 7.4 | 0.84 | 7.7 | -3.3 | 4.6 | 6.3 | 0.85 |
|  | PRD | 9.5 | 3.7 | -5.8 | 5.9 | 8.7 | 0.87 | 3.5 | -6 | 6.1 | 8.4 | 0.86 |

### 5.2.4 RH

Figure 9 shows the comparisons of observed and simulated RH. The spatial distribution of the MB of RH2m is influenced by the MB of T2m (the larger positive MB of T2m corresponds to the larger negative MB of RH2m), mainly because the calculation of RH is temperature dependent. For example, compared between these six regions, the MB of T2m in the CC region (1.4°C and 1.1°C for the NO-ACI$_{sub}$ and ACI$_{sub}$ experiment, respectively) is the largest, and thus the MB of RH2m (-17.1% and -14.8% for the NO-ACI$_{sub}$ and ACI$_{sub}$ experiment, respectively) is also the largest (Table 5). Compared between these two experiments, the ACI$_{sub}$ experiment generally has smaller MB of RH2m over this study area, with an overall 18.1% (relative changes) decrease in MB averaged over the six regions (from -13.8% to -11.3%) and an improvement in all other statistical indicators (Table 5), which suggests a better performance of the ACI$_{sub}$ experiment in RH2m predictions. For the vertical profile of RH, both the NO-ACI$_{sub}$ and ACI$_{sub}$ experiment have negative MB of RH below ~900 hPa and positive MB above ~900 hPa in most areas (Figure 9(f)). Due to the humidity-raising effects of the subgrid-scale ACI, the ACI$_{sub}$ experiment generally have a better performance than the NO-ACI$_{sub}$ experiment for RH at 1000-900 hPa in the study area, where the MAESS ranges of RH from 1000 to 900 hPa in the NEC, JJJ, CC, YRD, and PRD are 1% to 21%, 5% to 14%, 0.1% to 0.17%, 2% to 15%, and 7% to 13%, respectively (Figure. 11). A worsened performance of the RH simulation occurs at all air layers in the SB and above ~900 hPa in other regions due to an increase in the positive MB of the RH to some extent caused by subgrid-scale ACI, suggesting that the impact of subgrid-scale ACI on RH predictions also relates to the local errors of model itself.





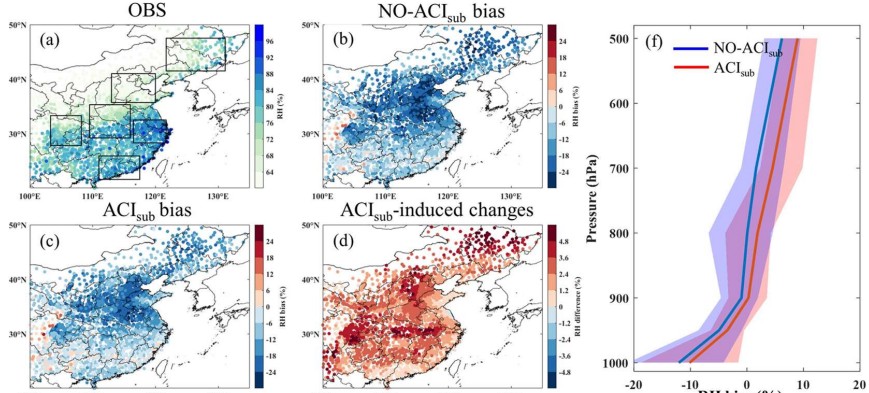

**Figure 9: The spatial distribution of time average RH2m and the vertical profiles of MB of RH in June 2016. (a) The observations.**

**(b) The MB of RH2m in the NO-ACI_sub experiment. (c) The MB of RH2m in the ACI_sub experiment. (c) The difference of RH2m**

**between the ACI_sub and NO-ACI_sub experiment. (f) The vertical profiles of MB of RH in the NO-ACI_sub and ACI_sub experiment. In**

**the (f), the shadings are the spread of MB of RH in six regions, and the solid lines are their average results.**

### 5.2.5 Wind speed

Figure 10 shows the comparisons of observed and simulated wind speed. The performance of the WS10m simulations

compared to observations is comparable to that of other studies and models (Table 5). Both NO-ACI_sub and ACI_sub experiments

overestimate WS10m over the study area, especially in the PRD, where the MB reaches 2.2 and 1.9 m s$^{-1}$, respectively. This

systematic overestimation of WS10m is a common problem in mesoscale NWP models, likely related to the treatment of the

underlying surface in the models (Jimenez and Dudhia, 2012; Jia and Zhang, 2021). For example, the complex underlying

surface of JJJ, YRD, and PRD cannot be fully resolved in this model, and the relatively smooth treatment of the underlying

surface leads to a significant overestimation of WS10m in these regions (Table 5). Compared with the NO-ACI_sub experiment,

the WS10m increases or decreases in different regions in the ACI_sub experiment and consequently increases or decreases the

MB, which leads to an overall less pronounced improvement in the MB of WS10m averaged over the six regions. As can be

seen from the other statistical indicators, the correlation coefficients of WS10m simulations for the different regions are

somewhat improved (Table 5). Further comparison reveals that the regions with increased WS10m are consistent with the

regions with significantly increased CLWP. It is speculated that it may be related to decreased atmospheric stability caused by

the more significant cooling in the upper atmosphere in these regions. In contrast, the decrease in WS10m is likely associated

with the increased atmospheric stability caused by the decline in the near-surface temperature. For the vertical profiles of wind

speed, both the NO-ACI_sub and ACI_sub experiments are in overall good agreement with observations. However, wind speed is

still overestimated in the lower air layers over most regions (Figure 10(f)). The comparison of the two experiments shows that

the subgrid-scale ACI has more complex effects on the vertical profile of wind speed than temperature or humidity, resulting

in an overall decrease in wind speed below ~800 hPa and an increase in wind speed above ~800 hPa. The MAESS values of

wind speed from 10 m to 500 hPa are also greater than 0 in most regions, reflecting the improvement of subgrid-scale ACI on



the vertical profile of wind speed. It is worth noting that this improvement varies significantly among different regions. For

example, the MAESS values over most air layers in the YRD and PRD are considerably larger than those in several other

regions (Figure 11), which may be related to cloud water content and local errors of model itself.

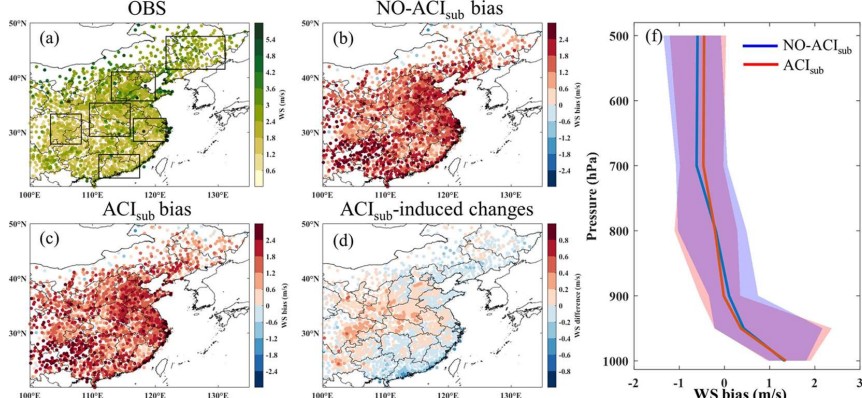

**Figure 10: The spatial distribution of time average WS10m and the vertical profiles of MB of wind speed in June 2016. (a) The observations. (b) The MB of WS10m in the NO-ACI_sub experiment. (c) The MB of WS10m in the ACI_sub experiment. (c) The difference of WS10m between the ACI_sub and NO-ACI_sub experiment. (f) The vertical profiles of MB of wind speed in the NO-ACI_sub**

**and ACI_sub experiment. In the (f), the shadings are the spread of MB of wind speed in six regions, and the solid lines are their average results.**

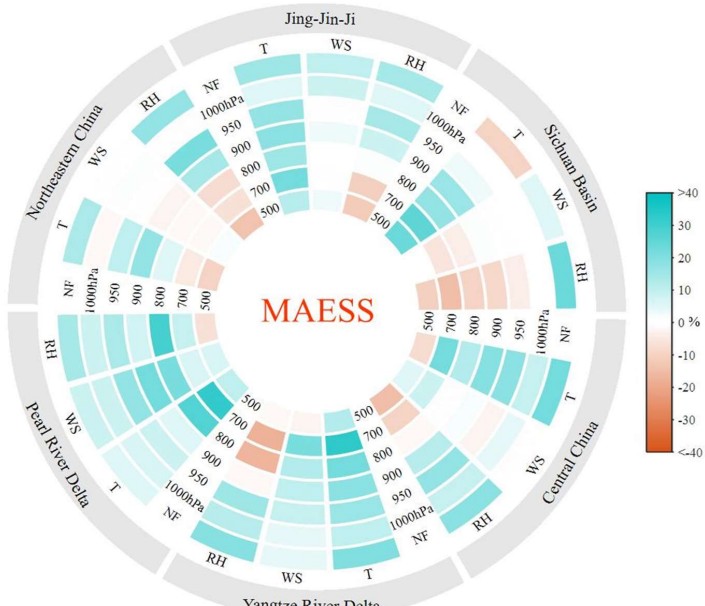

**Figure 11: Hourly MAESS (MAESS $= \left(1 - \frac{MAE_{ARI}}{MAE_{NO-ARI}}\right) \times 100\%$, where $MAE_{ARI}$ and $MAE_{NO-ARI}$ represent the mean absolute error**

**(MAE) (MAE $= |mean\ bias|$)) of predicted meteorological factors from the ACI_sub and NO-ACI_sub experiment in six regions (NEC,**

**JJ, SB, CC, YRD, and PRD). The green (red) filled boxes are the subgrid-scale ACI has positive (negative) effects.**

**5.2.6 Precipitation**



Figure 12 shows the comparisons of observed and simulated precipitation. Compared with observations, both NO-ACI$_{sub}$ and ACI$_{sub}$ experiments reproduce the overall spatial distribution of summer precipitation in the central and eastern China, with more in the south and less in the north. The related statistical indicators also show that the simulation performance of

precipitation is comparable to other models or studies (Table 5). The precipitation in the central and eastern China is significantly underestimated in the NO-ACI$_{sub}$ experiment, in which the MB of 24 hours cumulative precipitation in the NEC, JJJ, SC, CC, YRD, and PRD is -2, -1.8, -1.5, -3.1, -4.8, and -5.8 mm, respectively. Compared with the NO-ACI$_{sub}$ experiment, the 24 hours cumulative precipitation in the ACI$_{sub}$ experiment increases due to the significant enhancement of precipitation at grid-scale (figure omitted), which leads to an improvement in the underestimation of precipitation in the majority of regions,

where the MB of 24 hours cumulative precipitation in the NEC, JJJ, SC, CC, YRD, and PRD is -1.7, -1.5, 1.4, -1.5, -3.3, and -6 mm, respectively. Overall, the MB of 24 hours cumulative precipitation averaged over six regions decreased by 34.4% (from -3.2 to -2.1 mm). Previous studies have shown that explicit convective cloud microphysical processes at subgrid-scale significantly increase precipitation at grid-scale by enhancing the convective detrainment of cloud water and ice at a large scale (Song and Zhang, 2011), which is a key reason for the increase in precipitation in the ACI$_{sub}$ experiment. Other relevant

statistical indicators also show the improvement of 24 hours cumulative precipitation in the NEC, JJJ, SC, CC, and YRD by subgrid-scale ACI (Table 5). It is worth noting that the MB of precipitation in the PRD increases due to a slight decrease in precipitation, which is speculated to be related to the competing of water vapor among different regions (Glotfelty et al., 2020).

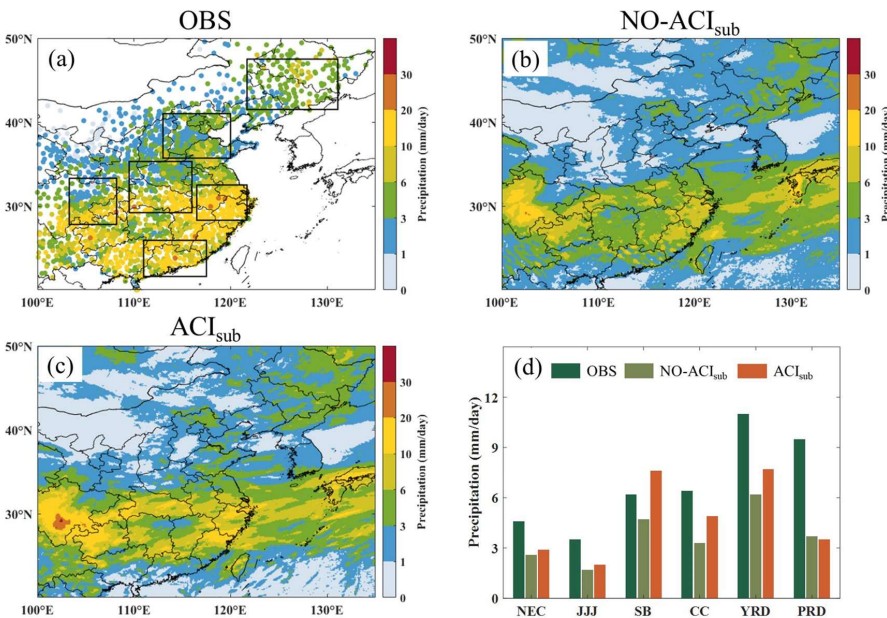

**Figure 12: The spatial distribution of time average 24 hours cumulative precipitation in June 2016 from the (a) observations, (b) NO-ACI$_{sub}$ experiment, and (c) ACI$_{sub}$ experiment. (d) The comparison of time average 24 hours cumulative precipitation in different regions.**



**5.3 Impact of anthropogenic aerosol on typical deep convective precipitation prediction via subgrid-scale ACI**

The discussion in the previous sections has shown that treating ACI at subgrid-scale in this model improves the performance

of most predicted meteorological factors. In this section, the model coupled with subgrid-scale ACI is utilized to separately
explore the effects of anthropogenic aerosol perturbations at subgrid-scale by controlling anthropogenic aerosol emissions for
a typical deep convective precipitation event.

The individual case chosen for the study is a continuous heavy precipitation event from 26 to 29 June 2016 in the YRD. During
this period, the YRD region is influenced by a deep convective cloud system (Figure 13), with the regionally averaged

cumulative precipitation approaching 90 mm (Figure 15 (a, b)), and the model can reproduce the precipitation event (Figure
15 (c)). As shown in Figure 13, on 26 June 2016, convective cloud with high cloud top pressure and low cloud top height is
over the YRD. On 27 and 28 June 2016, the cloud top pressure decreases and cloud top height rises, which is conducive to
water vapor condensation and precipitation production. As a result, the 24 hours cumulative precipitation exceeds 50 mm at
most stations during this period. On 29 June 2016, the convective cloud over this region gradually dissipates accompanied by

a decrease in precipitation. On 30 June 2016, the convective cloud completely dissipates. In addition, as shown in Figure 14,
the overall aerosol levels in the YRD are relatively low between 26 and 29 June 2016, with the peak of $PM_{2.5}$ mass
concentrations being less than 40 µg m$^{-3}$.

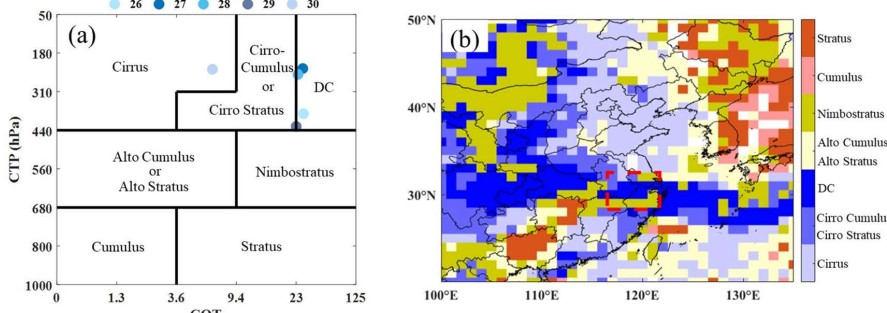

**Figure 13: (a) Cloud types over YRD from 26 to 30 June 2016 based on the International Satellite Cloud Climatology Project (ISCCP)
cloud classification algorithm (Hahn et al., 2001). (b) The spatial distribution of cloud types in central and eastern China on 28 June
2016. The red dashed rectangle is the location of the YRD region.**




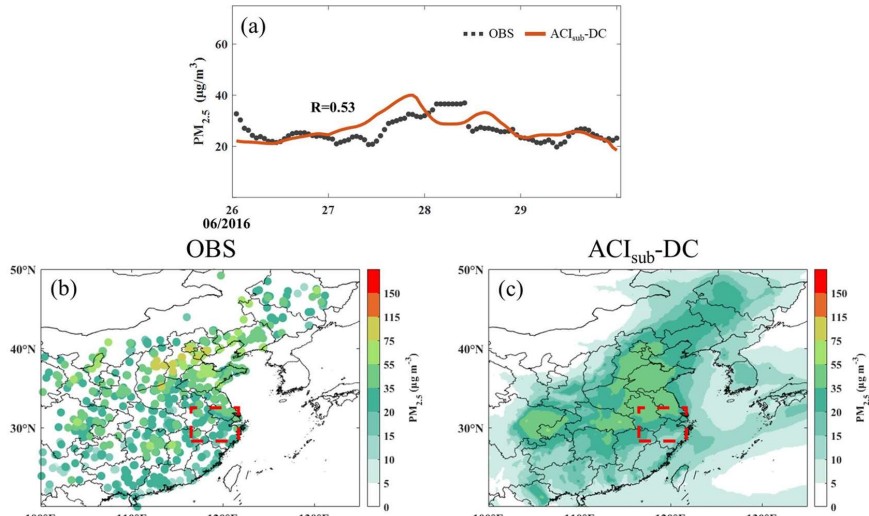

**Figure 14. (a) The temporal variation of regional average PM$_{2.5}$ mass concentration in YRD. The (b) spatial distribution of observed and (c) simulated by the ACI$_{sub}$-DC experiment time average PM$_{2.5}$ mass concentration from 26 to 29 June 2016**

Figure 15(d) shows the observed and simulated temporal variations of regional average hourly precipitation in the YRD. It can be seen that the simulations in both experiments are in good agreement with the observations, capturing both the rising and falling periods of precipitation, with R exceeding 0.7 (Figure 15(e)). The comparison of the ACI$_{sub}$-DC and CACI$_{sub}$-DC experiment shows that anthropogenic aerosol leads to a decrease in regional average precipitation in the YRD via subgrid-scale ACI, with a 5.6% (from 82 to 77.6 mm) decrease in cumulative precipitation for the study period. Compared with the CACI$_{sub}$-DC experiment, the ACI$_{sub}$-DC experiment shows a better performance in simulating this heavy precipitation event over the YRD, with R increasing from 0.7 to 0.73, centered root-mean-square-discrepancy (CRMSD) decreasing from 0.63 to 0.56, and standard deviation (STD) decreasing from 0.89 to 0.84 (Figure 15(e)).

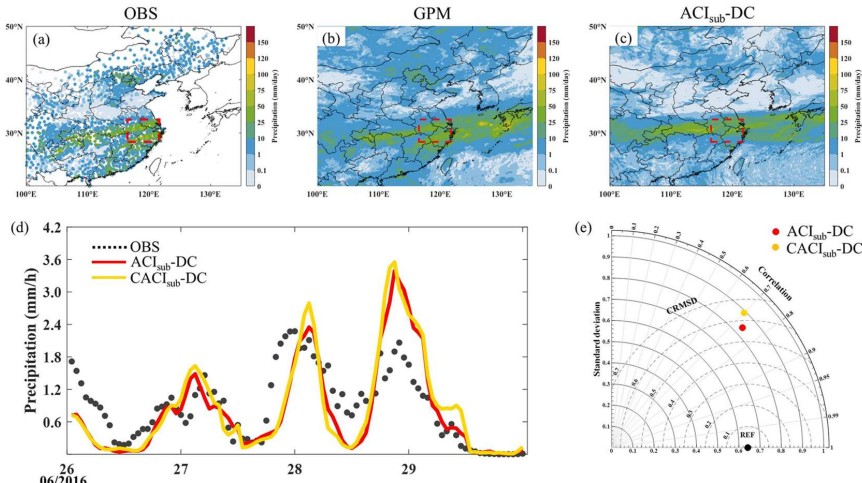

**Figure 15: The spatial distribution of time average 24 hours cumulative precipitation from 26 to 29 June 2016 in the (a) observations,**



**(b) GPM, and (c) ACI$_{sub}$-DC experiment. The (d) time variation and (e) Taylor diagram of observed and simulated regional average hourly precipitation in YRD from 26 to 29 June. In the Taylor diagram, the REF is the observation, the vertical coordinate is the standard deviation (STD), the distance between the simulations and REF is the centered root mean square deviation (CRMSD), and the position of the azimuth is the correlation coefficient (R).**

Further detailed analyses are carried out to investigate the causes of precipitation changes. Compared with the CACI$_{sub}$-DC experiment, the anthropogenic aerosol emissions in the ACI$_{sub}$-DC experiment leads to an increase in the average PM$_{2.5}$ mass concentration in the YRD during the study period by 23.5 μg m$^{-3}$ (Figure 16(a)), which directly causing the regional average cloud droplets number concentration of convective cloud at the subgrid-scale (averaged over 1-6 km) to increase by about 3.2×10$^6$ m$^{-3}$ (Figure 16(b)). Notably, the decreased cloud droplet number concentration within some YRD regions may be

related to changes in environmental supersaturation due to thermodynamic perturbations (Fan et al., 2016; Glotfelty et al., 2020). Anthropogenic aerosol directly induces the changes in cloud droplets number concentration at subgrid-scale, further influencing precipitation. The simulated precipitation is categorized into subgrid-scale precipitation from the cumulus convection scheme and grid-scale precipitation from the cloud microphysics scheme, and these two types of precipitation are studied separately. As can be seen in Figure 17 (a, c), the anthropogenic aerosol leads to a decrease in precipitation at both

subgrid-scale and grid-scale via subgrid-scale ACI, with the total cumulative precipitation during the study period decreasing by 2.9% (from 9.43 to 9.16 mm) and 5.9% (from 72.8 to 68.5 mm), respectively.

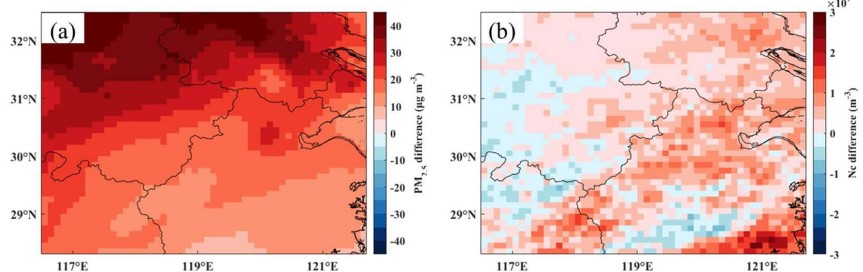

**Figure 16: The spatial distribution of the difference between the ACI$_{sub}$-DC and CACI$_{sub}$-DC experiment for the time average (a) PM$_{2.5}$ mass concentration and (b) subgrid-scale cloud droplets number concentration (mean values in 1-6 km) from 26 to 29 June**

**2016.**

The decrease in precipitation at subgrid-scale is mainly related to the weaker autoconversion of cloud water to rain at the subgrid-scale. As shown in Figure 17(b), it can be seen that there is a general increase in Qc (up to a maximum of 0.06 g kg$^{-1}$) at subgrid-scale in the ACI$_{sub}$-DC experiment compared to the CACI$_{sub}$-DC experiment. At the same time, the anthropogenic aerosol leads to the changes in Qc and radius of cloud droplets at subgrid-scale in the vertical direction showing a clear opposite

trend (Figure 18(a)). Based on this, it is reasonable to conclude that anthropogenic aerosol leads to more but smaller cloud droplets, which is unfavorable for the growth of cloud droplets into raindrops and inhibits the autoconversion process from cloud water to rainwater, thus leading to the increase of cloud water content and the decrease of precipitation at subgrid-scale. The combination of the location of the 0°C isotherm (a higher proportion of warm region in cloud) and the increase in Qi





(which usually leads to an increase in precipitation in the mixed-phase cloud dominated by cold cloud processes) roughly

excludes that anthropogenic aerosol leads to a decrease in precipitation at the subgrid-scale by influencing cold cloud processes

(Ma et al., 2015; Luo et al., 2023; Fan et al., 2016), which remains to be further analyzed in detail for precipitation sources

and sinks.

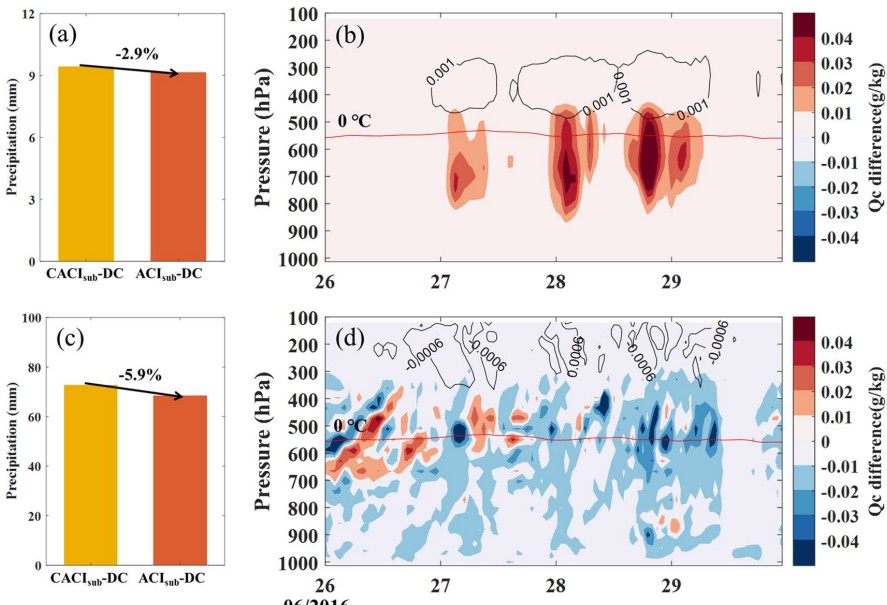

**Figure 17: The (top row) subgrid-scale and (bottom row) grid-scale (a and c) cumulative precipitation from 26 to 29 June 2016 and**

**vertical distributions of (b and d) difference between the ACI_sub-DC and CACI_sub-DC experiment for the regional average Qc and**

**Qi. In (b) and (d), the shading is Qc, the contour is Qi, and the red line is the 0°C isotherm.**

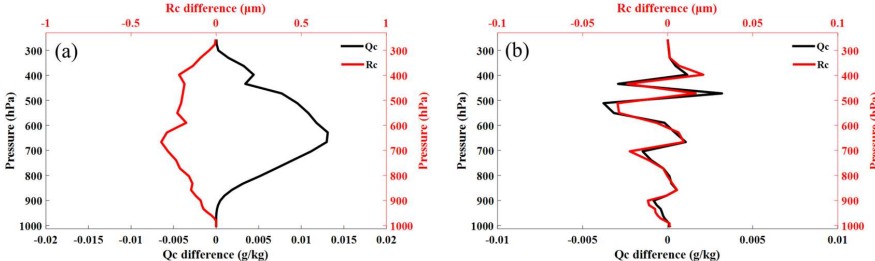

**Figure 18: (a) The difference of subgrid-scale Qc and Rc in YRD between the ACI_sub-DC and CACI_sub-DC experiment. (b) The**

**difference of grid-scale Qc and Rc in YRD between the ACI_sub-DC and CACI_sub-DC experiment.**

The decrease in precipitation at grid-scale is primarily related to competition of cloud at subgrid-scale for water vapor resulting

in less available water vapor for condensation at grid-scale. As shown in Figure 17(d), Qc at grid-scale decreases (up to a

maximum of -0.09 g kg$^{-1}$) over most air layers during the study period in the ACI_sub-DC experiment compared to the CACI_sub-

DC experiment. In contrast to the changes in the radius of cloud droplets at subgrid-scale, the changed trends of the radius of



cloud droplets and Qc at grid-scale in the vertical direction are the same (i.e., the radius of cloud droplets and cloud water

content decrease simultaneously) (Figure 18(b)). In addition, Qi, rain water mixing ratio (Qr), graupel mixing ratio (Qg), and

snow mixing ratio (Qs) decrease at grid-scale (Figure 19). These changes lead to a decrease in precipitation at grid-scale. Based

on the general reduction of all hydrometeors mixing ratio in cloud and smaller cloud droplets, it is reasonable to assume that

it is mainly related to the reduction of water vapor available for condensation at grid-scale. The anthropogenic aerosol-cloud

interaction at subgrid-scale is an important reason for the reduction of water vapor at grid-scale. Previous studies have also

shown a competing effect on water vapor between subgrid-scale and grid-scale cloud parameterization schemes, which is more

pronounced at subgrid-scale (Glotfelty et al., 2019, 2020).

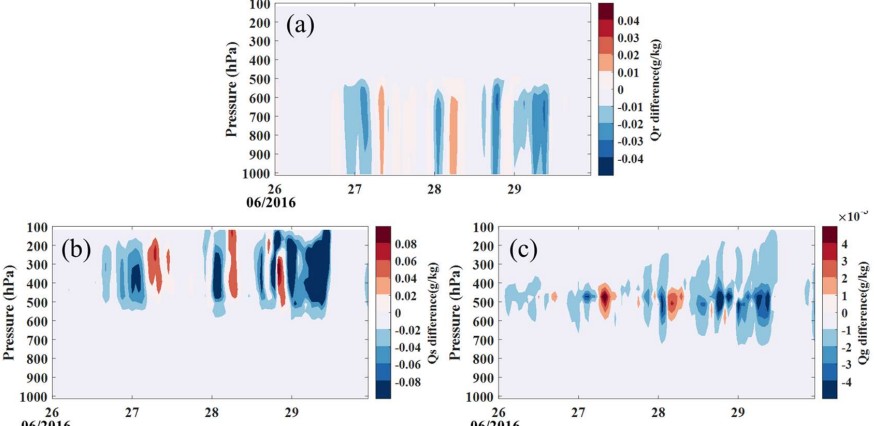

**Figure 19: The vertical distribution of difference between the ACIsub-DC and CACIsub-DC experiment for regional average grid-scale (a) Qr, (b) Qs, and (c) Qg in YRD from 26 to 29 June 2016.**

**6 Conclusions**

In this paper, based on an mesoscale atmospheric chemistry model CMA_Meso5.1/CUACE, the subgrid-scale ACI mechanism

is implemented for convective clouds with horizontal scales smaller than model grid spacing: a double-moment convective

cloud microphysical scheme (SZ2011), which explicitly deals with various hydrometeors (cloud water, cloud ice, rain, and

snow) microphysical processes of convective clouds, is coupled to the KFeta cumulus convective scheme; the real-time

predicted hygroscopic aerosol (OC, SS, SF, NT, and AM) by CUACE is used to generate cloud droplets at subgrid-scale via

ARG2000 size-resolved activation scheme; the calculated CF, Qc, Qi, Rc, and Ri in the KFeta cumulus convective scheme are

transferred to the Goddard shortwave radiation scheme for radiative feedback of subgrid-scale cloud. Based on reliable $PM_{2.5}$

mass and AOD simulations, two sets of experiments are conducted using this updated model. The first set of experiments

investigates the impact of the subgrid-scale ACI on the prediction of meteorological factors in summer in different regions

(NEC, JJJ, SC, CC, YRD, and PRD) of central and eastern China by whether or not to include the subgrid-scale ACI in the

model; the second set of experiments investigates the impact of anthropogenic aerosol on deep convective precipitation in the



YRD via subgrid-scale ACI.

The results show that the coupling of subgrid-scale ACI in the model refines cloud representations, e.g., causing underestimated cloud water content and cloud extinction to increase, even in some areas that are not saturated with respect to water at grid-

scale. As a result, the attenuation of shortwave radiation is better simulated with regional MB of SDSR decreasing by 23.1%. The cloud and radiation changes induced by subgrid-scale ACI lead to a decrease in temperature at 2 m accompanied by an increase in RH at 2 m, which helps to reduce regional MB by 40% and 18.1%, respectively. This cooling and humidification occur from 1000 hPa to 500 hPa, but the improvement is mainly concentrated in temperature at whole layers and RH below 900 hPa. Unlike temperature and RH, wind speed increases or decreases at different air layers or regions possibly related to

changes in atmospheric stability by subgrid-scale ACI. The subgrid-scale ACI further significantly enhances total precipitation at subgrid-scale and grid-scale, mainly causing by increased precipitation at grid-scale linked to convective detrainment, thus reducing regional MB of 24 hours cumulative precipitation by 34.4%. Compared with different subregions (NEC, JJJ, SCB, CC, YRD, and PRD) in central and eastern China, the subgrid-scale ACI effects on the prediction of meteorological factors is more significant in the YRD region, which is mainly related to convective conditions and model local errors. In addition,

compared with simulations with the anthropogenic emissions turned off, the subgrid-scale actual anthropogenic aerosol emissions make the grid-scale and subgrid-scale total cumulative precipitation during a typical deep convective heavy precipitation event in the YRD to decrease by 5.6% (4.6 mm), which is closer to the observations. It is further found that the decrease in total precipitation is associated with lower autoconversion of cloud water to rain at subgrid-scale and less water vapor available for condensation at grid-scale, suggesting the competing effect on water vapor between subgrid-scale and grid-

scale cloud.

The results of this study have pointed out the importance of the subgrid-scale ACI mechanism interacting with chemistry for the prediction of meteorological factors in NWP models and the necessity of multiscale ACI studies. However, there is still a need for some complementary work in the future, e.g., a study of the differences in the impact of the ACI mechanism on NWP at different grid resolutions (Glotfelty et al., 2020), the coupling of real-time ice crystals nucleation at grid-scale and subgrid-

scale and its impacts on the prediction of meteorological factors (Su and Fung, 2018a, b), the dependence of ACI effects on different cloud microphysics schemes and cumulus convection parameterization schemes (Miltenberger et al., 2018; Thompson and Eidhammer, 2014; Listowski and Lachlan-Cope, 2017; Zhang et al., 2021).

**Data availability**

The VIIRS daily Level-3 cloud and aerosol data are available at

https://ladsweb.modaps.eosdis.nasa.gov/archive/allData/5111/CLDPROP_D3_VIIRS_SNPP/2016/ and https://ladsweb.modaps.eosdis.nasa.gov/archive/allData/5200/AERDB_D3_VIIRS_SNPP/2016/. The CERES daily Level-3 radiation data are available at https://asdc.larc.nasa.gov/data/CERES/SYN1deg-Day/Terra-NPP_Edition1A/2016/06/. The GPM daily precipitation data are available at



https://gpm1.gesdisc.eosdis.nasa.gov/data/GPM_L3/GPM_3IMERGDF.07/2016/06/. The NCEP Final global analysis and
forecast data are available at https://rda.ucar.edu/datasets/ds083.3/.

**Author contributions**

Conceptualization: Hong Wang and Xiaoye Zhang. Methodology: Wenjie Zhang, Yue Peng, Zhaodong Liu, and Wenxing jia.
Investigation and Writing: Wenjie Zhang. Data curation: Junting Zhong and Deying Wang, Da Zhang. Validation: Chen Han,
Yang Zhao, and Huiqiong Ning. Supervision: Hong Wang, Xiaoye Zhang, and Huizheng Che.

**Competing interests**

The authors declare that they have no conflict of interest.

**Acknowledgments**

This study is supported by the NSFC Major Project (42090031); the National Key Research and Development Program of
China (2023YFC3706304).

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
