# Peer review of "Investigating the impact of subgrid-scale aerosol-cloud interaction on mesoscale meteorology prediction"

_EGUsphere, 2024_

## Referee Comment (RC1)

**Review of "Investigating the impact of subgrid-scale aerosol-cloud interaction on mesoscale meteorology prediction" by Zhang et al., submitted to Atmospheric Chemistry and Physics (ACP)**

[Article#: acp-2024-3677]

This report contains general, major, and specific comments from this reviewer on the manuscript.

**A summary of the manuscript and general assessment:**

Recommendation: Major Revision

This study implemented an update of the Kain and Fritsch subgrid convection parameterization by incorporating cloud microphysics with aerosol-cloud interaction (ACI) into a mesoscale atmospheric chemistry model, CMA_Meso5.1/CUACE. The performance of the updated model was evaluated in two types of tests. The first test compared model simulations with and without the update of the subgrid convection parameterization, and investigated the differences by comparison with satellite and ground-based observations for June 2006. Overall, the update reduced various model biases, primarily through improved representation of the modeled cloud for the atmospheric radiation calculation. The second test configured simulations with and without anthropogenic emissions for several days in late June, when heavy surface precipitation was observed over southern China. The inclusion of anthropogenic emissions resulted in a reduction of surface precipitation due to a lower auto-conversion rate of cloud water to rain and less water vapor available for grid-scale condensation.

This study is within the aims and scope of Atmospheric Chemistry and Physics (ACP), specifically the subject for "Aerosols, Cloud and Precipitation", the research activity for "Atmospheric modelling", the altitude range for "Troposphere", and the science focus on "both Chemistry and Physics".

As mentioned in the text, the subgrid ACI effects are often overlooked in the modeling studies for ACI. It is scientifically significant to investigate this problem by implementing the effects on the subgrid convection parameterization and conducting simulations. The direction and approach of the research is reasonable and acceptable. However, I think that the current manuscript contains a number of misinterpretations of the results and major misleading descriptions, especially for the first set of simulations. These issues should be adequately addressed and corrected before

acceptance for publication. Detailed comments are provided below.

**Major comments:**

1. Literature review of previous studies for modeling subgrid ACI effects

The introduction section needs to include more reviews of previous studies for aerosol-aware sub-grid convective parameterization. For example, Grell and Freitas (2014) is a widely used and, to my knowledge, the most cited work for sub-grid ACI parameterization, although the approaches for microphysical representation are very coarse compared to Song and Zhang (2011) and Glotfelty et al. (2019, 2020). It would be better to describe what is new and novel compared to these previous studies, in order to highlight the significance of this study.

Grell, G. A., & Freitas, S. R. (2014). A scale and aerosol aware stochastic convective parameterization for weather and air quality modeling. Atmospheric Chemistry and Physics, 14(10), 5233-5250.

2. Parameterizing sub-grid updraft (or sub-grid supersaturation) for ARG2000

The current descriptions of subgrid ACI parameterization are missing important information, especially how to parameterize the subgrid updraft and its variability that needs to be entered into $\zeta$ and $\eta$ in the ARG2000 scheme. I think this is the most difficult part of implementing cloud microphysics, considering the effects of aerosol nucleation on cloud droplets, such as the ARG2000 scheme, into subgrid convective parameterization. On line 183, "Meteorological factors include atmospheric vertical velocity, temperature, etc., which can be provided in real time by the CMA_Meso5.1 model". Since the grid-scale vertical velocity cannot be used here as the subgrid-scale vertical velocity for the ARG2000 scheme, the subgrid-scale vertical velocity needs to be prepared somehow. Song and Zhang (2011) and Glotfelty et al. (2019) use different approaches to parameterize the subgrid scale vertical velocity. Please clarify how to parameterize the subgrid-scale vertical velocity in this study, and add detailed descriptions in the text.

3. VIIRS AOD comparison (Section 5.1 and Figure 4)

I think there are possible misinterpretations of the VIIRS AOD data. First of all, the VIIRS AOD is "clear sky" AOD because COT is generally much higher than AOD, so retrieval algorithms for typical space-borne radiometers cannot calculate "cloudy sky" AOD. Thus, my first question is whether the simulation AOD in Fig. 4d is really clear-sky AOD or all-sky AOD. If the simulation AOD is all-sky AOD, then it causes an underprediction because cloudy sky AOD could be lower

than clear sky AOD due to wet scavenging by precipitation. Second, I cannot believe that the real clear-sky AOD over South China is too low, such as 0 ~ 0.04, as shown in Fig. 4c. This strangely too low AOD is clearly inconsistent with the surface PM2.5 data in Fig. 4a as well as other observational data, such as the MODIS AOD climatology shown below. I think that clear sky AOD cannot be calculated from the satellite observations over the region for that month, because the region was covered by clouds on almost all days, as shown in Fig. 5a. Thus, I just wonder if the actual VIIRS AOD is "undefined" rather than 0 or really low values. Please check the downloaded data products and the process for plotting.

[Figure]

Adapted from, Ratnam, M.V., Prasad, P., Raj, S.T.A. et al. Changing patterns in aerosol vertical distribution over South and East Asia. Sci Rep 11, 308 (2021). https://doi.org/10.1038/s41598-020-79361-4

4. NO-ACIsub vs. ACIsub
This is the most important problem I ask the authors to address. As long as I read the whole section 5.2 for the first set of experiments, I think the drastic changes in the simulation results between NO-ACIsub vs. ACIsub (Table 3) come from the inclusion of the subgrid-scale cloud in the calculation of the atmospheric radiation processes, rather than from the inclusion of the aerosol effect for the subgrid-scale cloud microphysics. Therefore, I feel that the current descriptions of the difference between the two experiments, such as Table 3, may be misleading or exaggerated. If the authors want to show and discuss the result changes in cases with and

without the subgrid-scale ACI effects, the results should be presented in a way that disentangles the two components, the inclusion of the subgrid-scale cloud in the calculation of the atmospheric radiation processes and the inclusion of the aerosol effect for the subgrid-scale cloud microphysics. I am aware that SZ2011 eventually uses ARG2000. However, this problem should be critically addressed because it is the core of the research topic and goal.

5. CERES data comparison in 5.2.2

I am really confused with what is going on in this subsection. Please redo the work. The CERES data products provide the top-of-atmosphere (TOA) upward radiation fluxes, not the surface downward radiation fluxes (how satellite sensors can directly measure the surface downward radiation fluxes...). Thus, Figs. 6a and 6d should show the upward TOA shortwave and longwave radiation fluxes. I have no idea which TOA upward or surface downward fluxes from the simulations are shown for the rest of the panels. The selections of the color table and contour ranges of Fig. 6 are quite messy, which further hinders my understanding.

6. Sampling timing of the simulation results for comparison with the daily products from polar orbiting satellites

The VIIRS and CERES sensors on the SNPP satellite measures a location only twice (daytime and nighttime) per day due to the polar-orbiting so that their daily products are based on the observed values at specific local time (daytime only or both) within a day. I wonder if the authors actually sampled the simulation results for the comparison at specific timing on the days as much as similar to the satellite flying timing. This is often important, especially for validation of cloud, because cloud and precipitation lifecycles have a strong diurnal cycle in summer as shown in Fig. 15d.

7. R difference between ACIsub-DC and CACIsub-DC in Section 5.3

In Section 5.3, the explanation of the mean bias of surface precipitation sounds reasonable. However, I am not convinced how the authors argue that R is also improved. The 0.03 between 0.7 and 0.73 of R is, in my opinion, almost the same or kind a level of random error noise. If the authors want to argue the improvement of R, please add some follow-up descriptions on the mechanism for improving R.

**Specific comments:**

Abstract: Please refine the abstract to help readers understand the conclusions of the study, rather

than just listing the result changes in % values.

Line 158: The equation looks to be missing some components.

Line 228: "with a forecast time of 24 hours", does this mean a 24 hour forecast loop similar to Zhang et al. (2022)?

Figure 6: Please change color map and scale.

Line 315: Please clarify that the comparison with surface (ground-based) station data for SDSR starts from here.

Line 415: "The related statistical indicators also show that the simulation performance of precipitation is comparable to other models or studies (Table 5)." I do not understand what is meant here, especially "other models or studies". Please clarify.

Line 474: "Notably, the decreased cloud droplet number concentration within some YRD regions may be related to changes in environmental supersaturation due to thermodynamic perturbations (Fan et al., 2016; Glotfelty et al., 2020)." I do not understand what is meant here. Please specify which parts of the two publications I should read to understand.

**Grammatical problems:**

Abbreviations are sometimes not fully spelled out the first time they appear. Please check again.

---

## Author Comment (AC1)

Dear Reviewer:

Thank you for careful comments. These comments are all valuable and very helpful for revising and improving our paper, as well as the important guiding significance to our researches. We have studied the comments carefully and have made corrections which we hope meet with approval. The Reviewer's comments are in blue and our responses are in black. Revised portion are marked in red in the marked-up manuscript. The main corrections in the paper and the point-to-point responses are as following:

Response to Reviewer #1:

Main comments:

**1. Literature review of previous studies for modeling subgrid ACI effects. The introduction section needs to include more reviews of previous studies for aerosol-awaresub-grid convective parameterization. For example, Grell and Freitas (2014) is a widely used and,to my knowledge, the most cited work for sub-grid ACI parameterization, although theapproaches for microphysical representation are very coarse compared to Song and Zhang (2011)and Glotfelty et al. (2019, 2020). It would be better to describe what is new and novel compared to these previous studies, in order to highlight the significance of this study.**

Response: Accept. We have added more reviews of studies on subgrid-scale ACI in the introduction section, including studies of Grell and Freitas (2014), Lohmann (2008), Song and Zhang (2011), Lim et al. (2014), Glotfelty et al. (2019). Relevant contents have been added to the revised manuscript in lines 80-90.

Additionally, we emphasize the uniqueness and innovativeness of this study compared to these previous studies. In terms of model development, this study completes the full coupling of the double-moment cumulus scheme with the aerosol module and the short-wave radiation scheme in the atmospheric chemistry model, which makes the emissions-aerosol-subgrid-scale cloud-radiation/precipitation processes are closed, allowing the effects of subgrid-scale ACI on radiation and precipitation to be

investigated in more realistic aerosol levels; in terms of model application, this study more systematically evaluates the impacts of the treatment of subgrid-scale cloud microphysics and radiation feedback on multiple predicted meteorological factors, rather than limiting them to cloud and precipitation. Relevant contents have been added to the revised manuscript in lines 102-105.

**2. Parameterizing sub-grid updraft (or sub-grid supersaturation) for ARG2000The current descriptions of subgrid ACI parameterization are missing important information,especially how to parameterize the subgrid updraft and its variability that needs to be entered into $\zeta$ and $\eta$ in the ARG2000 scheme. I think this is the most difficult part of implementing cloudmicrophysics, considering the effects of aerosol nucleation on cloud droplets, such as theARG2000 scheme, into subgrid convective parameterization. On line 183, "Meteorologicalfactors include atmospheric vertical velocity, temperature, etc., which can be provided in realtime by the CMA_Meso5.1 model". Since the grid-scale vertical velocity cannot be used here asthe subgrid-scale vertical velocity for the ARG2000 scheme, the subgrid-scale vertical velocityneeds to be prepared somehow. Song and Zhang (2011) and Glotfelty et al. (2019) use different approaches to parameterize the subgrid scale vertical velocity. Please clarify how to parameterizethe subgrid-scale vertical velocity in this study, and add detailed descriptions in the text.**

Response: Accept. We have added the description of the parameterization of subgrid-scale vertical velocity in lines 221-235 in the revised manuscript, the details of which are shown as follows:

The subgrid-scale vertical velocity ($w_{sub}$) is determined by the updraft kinetic energy ($K_{sub}$):

$$w_{sub} = \sqrt{2K_{sub}} \qquad (5)$$

$$\frac{\partial K_{sub}}{\partial z} = -\frac{v_w}{M_w}(1 + \beta C_d)K_{sub} + \frac{1}{f(1+\lambda)}g\frac{T_{wu}-T_{we}}{T_{wu}} \qquad (6)$$

$$T_{wu} = T_u(1 + 0.608Qu - Qr - Qi - Qc - Qs) \qquad (7)$$

$$T_{we} = T_e(1 + 0.608Qe) \qquad (8)$$

where $v_w$ is the larger of entrainment or detrainment mass flux and $M_w$ is the convective updraft mass flux in the Kfeta scheme. The $\beta$, $C_d$, $\lambda$, and $f$ are constants, which are set to 1.875, 0.506, 0.5, and 2. The g is gravitational acceleration. $T_{wu}$ and $T_{we}$ are the density

temperature of updraft and environment, which can be solved by equations (7) and (8). In equation (7), $T_u$ is the temperature of updraft, Qu is the specific humidity of updraft, and Qr (Qi, Qc, or Qs) is the rain (ice, cloud, or snow) water mixing ratio. In equation (8), $T_e$ is the temperature of environment and Qe is the specific humidity of environment. The calculation of subgrid-scale vertical velocity refers to the method in Section 2.2 of the study by Song and Zhang (2011). The minimum value of the subgrid-scale vertical velocity is set to 0.5 m s$^{-1}$ at the cloud base and the maximum value is less than 20 m s$^{-1}$.

**3. VIIRS AOD comparison (Section 5.1 and Figure 4)I think there are possible misinterpretations of the VIIRS AOD data. First of all, the VIIRS AOD is "clear sky" AOD because COT is generally much higher than AOD, so retrieval algorithms fortypical space-borne radiometers cannot calculate "cloudy sky" AOD. Thus, my first question is whether the simulation AOD in Fig. 4d is really clear-sky AOD or all-sky AOD. If the simulation AOD is all-sky AOD, then it causes an underprediction because cloudy sky AOD could be lower than clear sky AOD due to wet scavenging by precipitation. Second, I cannot believe that the realclear-sky AOD over South China is too low, such as 0 ~ 0.04, as shown in Fig. 4c. This strangelytoo low AOD is clearly inconsistent with the surface PM2.5 data in Fig. 4a as well as otherobservational data, such as the MODIS AOD climatology shown below. I think that clear skyAOD cannot be calculated from the satellite observations over the region for that month, becausethe region was covered by clouds on almost all days, as shown in Fig. 5a. Thus, I just wonder ifthe actual VIIRS AOD is "undefined" rather than 0 or really low values. Please check thedownloaded data products and the process for plotting.**

Response: Revised. We remove the comparison between the simulated AOD and the VIIRS AOD data in the manuscript because the simulated AOD is the all-sky AOD as it includes contributions from both clear and cloudy conditions, whereas the VIIRS AOD mainly represents the clear sky AOD. These two data do not match exactly, especially in South China where there is a large amount of missing AOD data due to cloud cover, which also leads to AOD is too low.

Instead, we added a comparison of the simulated AOD with the MERRA2 AOD data (which provides all-sky AOD data) in order to evaluate the model aerosol simulation performance in lines 308-316 in the revised manuscript, the details of which are shown as follows:

The MERRA-2 data show that the regional average AOD is 0.42, 0.62, 0.35, 0.50, 0.52, and 0.27 in the NEC, JJJ, SB, CC, YRD, and PRD, respectively. The CMA_Meso5.1/CUACE model seems to capture some high-value and low-value areas of AOD well in the south of the domain (e.g., the regional average AOD is 0.31, 0.41, and 0.20 in the SB, YRD, and PRD with MB of -0.04, -0.11, and -0.07) but significantly underestimates AOD in the north of the domain (e.g., the regional average AOD is 0.14, 0.28, and 0.32 in the NEC, JJJ, and CC with MB of -0.28, -0.34, and -0.18). This substantially underestimated AOD in the NEC and JJJ region accompanied by underestimated $PM_{2.5}$ mass concentration is possibly related to underestimated anthropogenic emissions, inadequate representation of aerosol chemical reaction processes, etc. Compared with other studies or models, the CMA_Meso5.1/CUACE model has a similar performance in predicting AOD over China in summer (Werner et al., 2019; Wang et al., 2021; He et al., 2022).

[Figure]

**Figure 4: Spatial distribution of time average $PM_{2.5}$ (a and b) and AOD (c and d) in June 2016 from the NO-**

**ACIsub experiment compared against the observations and MERRA-2 data.**

**4. NO-ACIsub vs. ACIsubThis is the most important problem I ask the authors to address. As long as I read the whole section 5.2 for the first set of experiments, I think the drastic changes in the simulation results between NO-ACIsub vs. ACIsub (Table 3) come from the inclusion of the subgrid-scale cloud in the calculation of the atmospheric radiation processes, rather than from the inclusion of the aerosol effect for the subgrid-scale cloud microphysics. Therefore, I feel that the current descriptions of the difference between the two experiments, such as Table 3, may be misleading or exaggerated. If the authors want to show and discuss the result changes in cases with and without the subgrid-scale ACI effects, the results should be presented in a way that disentangles the two components, the inclusion of the subgrid-scale cloud in the calculation of the atmospheric radiation processes and the inclusion of the aerosol effect for the subgrid-scale cloudmicrophysics. I am aware that SZ2011 eventually uses ARG2000. However, this problem should be critically addressed because it is the core of the research topic and goal.**

Response: Revised. Thank you for your suggestions. Firstly, we have modified the description of the first set of experiments, in particular Table 3 and Section 5.2. The whole Section 5.2 for the first set of experiments represents the impact of model development (subgrid-scale cloud microphysics and radiation feedback in this model) on the prediction of meteorological factors. This is important for the investigation of subgrid-scale ACI, but our presentation may have led to misunderstandings, especially the name of the experiment. especially the name of the experiment. Therefore, we make a clearer distinction between the model development (subgrid-scale cloud microphysics and radiation feedback in this model) and the subgrid-scale ACI effects in the revised manuscript, which mainly includes that modifying the names and descriptions of the experiments throughout the manuscript, and removing descriptions about the impact of subgrid-scale ACI in Section 5.2. In addition, corresponding changes are made in the Abstract, Introduction, Model configurations and experimental design, and other analysis of results and conclusions.

**Table 3: Descriptions of multiple sensitivity experiments.**

| Experiment | Description |
| --- | --- |
| CONTROL | Model runs without subgrid-scale cloud microphysics and cloud radiation feedback |
| CU-MP-RA | Same as CONTROL, but with subgrid-scale cloud microphysics and cloud radiation feedback |
| ACI$_{sub}$-DC | Same as CU-MP-RA, but for a deep convective process and fixing the cloud droplets number concentration in the Thompson cloud microphysics scheme as 300 cm$^{-3}$ |
| CACI$_{sub}$-DC | Same as ACI$_{sub}$-DC, but turning off MEIC anthropogenic emissions |

Secondly, we have redefined the objective and research contents of this study in lines 97-107 in the revised manuscript. The overall goal of this study is to achieve quantifiable subgrid-scale ACI in the atmospheric chemistry model CMA_Meso5.1/CUACE and to understand the impact of subgrid-scale ACI on numerical weather prediction (NWP). Based on this, we introduce the development of the model for implementing subgrid-scale ACI mechanism in the model in Section 3, evaluate the preformance of the developed model with subgrid-scale cloud microphysics and radiation feedback in Section 5.2, and investigate the impact of anthropogenic aerosol on typical deep convective precipitation prediction via subgrid-scale ACI in Section 5.3.

As you mentioned, assessing the relative contributions of subgrid-scale cloud microphysics and radiation feedback is another critical research focus, despite the inherent complexity of reasons. For example, Lim et al. (2014) show that subgrid-scale cloud microphysics in the WRF model improves overestimated radiation and underestimated precipitation during the East Asian monsoon season by increasing detrained cloud water and cloud ice; Alapaty et al. (2012) find that introducing subgrid-scale cloud radiation feedback for regional meteorological and climate modeling makes the attenuation of SDSR more realistic and further suppressed convection. Future detailed discussions will continue in the next paper, including systematically distinguishing the differences between subgrid-scale cloud microphysics and radiation feedback effects on meteorological prediction, as well as quantifying the subgrid-scale ACI effects in detail through controlled emission experiments. We have added these contents to the Conclusion in lines 651-654 in the revised manuscript.

**5. Sampling timing of the simulation results for comparison with the daily products from polar orbiting satellites. The VIIRS and CERES sensors on the SNPP satellite measures a location only twice (daytime and nighttime) per day due to the polar-orbiting so that their daily products are based on the observed values at specific local time (daytime only or both) within a day. I wonder if the authors actually sampled the simulation results for the comparison at specific timing on the days as much as similar to the satellite flying timing. This is often important, especially for validation of cloud, because cloud and precipitation lifecycles have a strong diurnal cycle in summer as shown in Fig.15d.**

Response: Yes. We actually sample and calibrate the model simulations based on the transit times of the VIIRS and CERES satellites, and the relevant explanations have been added to the revised manuscript in lines 335-338 and 374-375, the details of which are shown as follows:

The daily cloud properties data from VIIRS used in this study consist solely of visible-band products, which are available only during local daytime. For comparative evaluation, the model simulations are sampled according to transit times of satellites over China. The transit time of VIIRS over China occurs approximately between 13:00 and 14:00 local time, and the corresponding simulations for comparison are averaged hourly data at 13:00 and 14:00 local time.

The daily radiation properties from CERES are computed with hourly data derived from MODIS and geostationary satellites (GEO), and the corresponding simulations for comparison are 24-hour averaged values.

**6. R difference between ACIsub-DC and CACIsub-DC in Section 5.3 In Section 5.3, the explanation of the mean bias of surface precipitation sounds reasonable.However, I am not convinced how the authors argue that R is also improved. The 0.03 between0.7 and 0.73 of R is, in my opinion, almost the same or kind a level of random error noise. If theauthors want to argue the improvement of R, please add some follow-up descriptions on themechanism for improving R.**

Response: Revised. Thank you for highlighting the need to clarify the R value

improvement. We agree that R from 0.7 to 0.73 does not represent an improvement of R and remove the related contents in lines 557 in the revised manuscript.

Specific comments:

**1. Abstract: Please refine the abstract to help readers understand the conclusions of the study, ratherthan just listing the result changes in % values.**

Response: Revised. We have refined the abstract in the revised manuscript.

**2. Line 158: The equation looks to be missing some components.**

Response: Revised. It is possible that the PDF is generated with a missing T (ambient temperature). We have regenerated the PDF and the equation can be displayed normally.

**3. Line 228: "with a forecast time of 24 hours", does this mean a 24 hour forecast loop similar to Zhang et al. (2022)?**

Response: Yes. The "with a forecast time of 24 hours" means "a 24-hour forecast loop" similar to Zhang et al. (2022)"

**4. Figure 6: Please change color map and scale.**

Response: Accept. We have redrawn the Figure 6 by using different color map and scale.

[Figure]

**Figure 6: The spatial distribution of time average (a-c) SDSR, (d-f) SDLR in June 2016. The left, middle, and right column is the CERES, CONTROL, and CU-MP-RA experiment, respectively.**

**5. Line 315: Please clarify that the comparison with surface (ground-based) station data for SDSR starts from here.**

Response: Accept. We have added relevant explanations to clarify the comparison with surface (ground-based) station data for SDSR in lines 388-389 in the revised manuscript.

**6. Line 415: "The related statistical indicators also show that the simulation performance of precipitation is comparable to other models or studies (Table 5)." I do not understand what ismeant here, especially "other models or studies". Please clarify.**

Response: Revised. The sentence "The related statistical indicators also show that the model's simulation performance of precipitation is comparable to other models or studies (Table 5)." is changed to "The values of related statistical indicators (Table 5) also show that the simulation performance of precipitation is similar to that of other NWP models (e.g., WRF-CMAQ, WRF, etc.) or results reported in previous studies (Glotfelty et al., 2019; Wang et al, 2021; Wong et al. 2012)." in lines 502-504 in the revised manuscript.

**7. Line 474: "Notably, the decreased cloud droplet number concentration within some YRD regions may be related to changes in environmental supersaturation due to thermodynamic perturbations (Fan et al., 2016; Glotfelty et al., 2020)." I do not understand what is meant here. Please specify which parts of the two publications I should read to understand.**

Response. Revised. The sentence means that we find that aerosol mass concentration in the YRD region increase overall, but cloud droplet number concentration does not increase accordingly, and decreases in some areas. According to the section 2 about the impact of CCN effects on deep convective clouds of the study by Fan et al. (2016) and the section 2 about the changed cloud liquid water induced by decreasing aerosol concentration of the study by Glotfelty et al. (2020), we think that the decreased cloud droplet number concentration may be related to lower environmental supersaturation due to thermodynamic/dynamic perturbations (e.g. weaker updrafts, evaporative cooling). These revisions are in lines 570-571 in the revised manuscript.

Grammatical problems:

**8. Abbreviations are sometimes not fully spelled out the first time they appear. Please check again.**

Response: Revised. We have rechecked all the abbreviations throughout the manuscript and revised them accordingly.

---

## Author Comment (AC2)

Dear Reviewer:

Thank you for careful comments. These comments are all valuable and very helpful for revising and improving our paper, as well as the important guiding significance to our researches. We have studied the comments carefully and have made corrections which we hope meet with approval. The Reviewer's comments are in blue and our responses are in black. Revised portion are marked in red in the marked-up manuscript. The main corrections in the paper and the point-to-point responses are as following:

Response to Reviewer #2:

Main comments:

**1). Model Configuration and Experimental Design: The authors claim that the difference between the NO-ACIsub and ACIsub runs represents the impact of subgrid-scale ACI in the first set of experiments. However, the control run (NO-ACIsub) uses the regular KF cumulus parameterization without microphysical processes in convection, while the ACIsub run incorporates both subgrid-scale microphysics and subgrid-scale aerosol-cloud interactions. In the title, abstract, results analysis, and conclusion, the authors attribute the differences between the ACIsub and NO-ACIsub runs solely to subgrid-scale ACI. However, some of the significant differences could be due to the subgrid-scale microphysics. This could mislead readers into attributing all the differences to subgrid-scale aerosol-cloud interactions. The authors should reconsider the experimental design to more clearly distinguish the effects of subgrid-scale ACI.**

Response: Accept. We have reworked the description of the first set of experiments design to more clearly distinguish between the model development (coupling subgrid-scale cloud microphysics and radiation feedback) and the subgrid-scale ACI effects throughout the revised manuscript. The detailed modifications are described below:

Throughout the manuscript, the names of the first set of experiments are modified. The NO-ACI$_{sub}$ experiment is renamed the CONTROL experiment to represent the model

runs without subgrid-scale cloud microphysics and cloud radiation feedback. The ACI$_{sub}$ experiment is renamed the CU-MP-RA experiment to represent the model runs with subgrid-scale cloud microphysics and cloud radiation feedback.

In the **Abstract**, the original contents have been modified as following:

Aerosol-cloud interaction (ACI) significantly influences global and regional weather and is a critical focus in numerical weather prediction (NWP), but subgrid-scale ACI effects are often overlooked. Here, subgrid-scale ACI mechanism is implemented by explicitly treating cloud microphysics in KFeta convective scheme with real-time size-resolved hygroscopic aerosol activation, and introducing subgrid-scale cloud radiation feedback in an atmospheric chemistry model CMA_Meso5.1/CUACE. Focus on summer over central and eastern China, the performance evaluation shows that this developed model with subgrid-scale cloud microphysics and radiation feedback refines cloud representation even in some grid-scale unsaturated areas and subsequently leads to attenuated surface downward shortwave radiation (~18.5 W m$^{-2}$) more realistic. The increased cloud radiative forcing results in lower temperature (~0.35$^{\circ}$C) and higher relative humidity (~2.5%) at 2 m with regional mean bias (MB) decreasing by ~40% and ~18.1%. Temperature vertical structure and relative humidity below ~900 hPa are improved accordingly due to cooling and humidifying. The underestamated precipitation is enhanced, especially at grid-scale, thus reducing regional MB by ~34.4% (~1.1 mm). The performance differences between various subregions are related to convective conditions and model local errors. Additionally, compared to simulations with anthropogenic emissions turned off, subgrid-scale actual aerosol inhibits cumulative precipitation during a typical heavy rainfall event by ~4.6 mm, aligning it with observations, associated with lower autoconversion at subgrid-scale and less available water vapor for grid-scale condensation, suggesting competitions between subgrid- and grid-scale cloud. This study contributes to the understanding of the impact of subgrid-scale ACI on NWP.

In the **Introduction**, we have redefined the objective and research contents of this study in lines 97-107 in the revised manuscript.

In the **Model configurations and experimental design,** we have rewrote the descriptions of multiple sensitivity experiment in lines and Table 3 in the revised manuscript.

Table 3: Descriptions of multiple sensitivity experiments.

| Experiment | Description |
|---|---|
| CONTROL | Model runs without subgrid-scale cloud microphysics and cloud radiation feedback |
| CU-MP-RA | Same as CONTROL, but with subgrid-scale cloud microphysics and cloud radiation feedback |
| ACI$_{sub}$-DC | Same as CU-MP-RA, but for a deep convective process and fixing the cloud droplets number concentration in the Thompson cloud microphysics scheme as 300 cm$^{-3}$ |
| CACI$_{sub}$-DC | Same as ACI$_{sub}$-DC, but turning off MEIC anthropogenic emissions |

In the **Results and discussions**, we have reworked the contents about performance evaluation for the first set of experiments, especially in Section 5.2, where we have revised the section title to "Performance evaluation of predicted meteorological factors", and deleted the description of subgrid-scale ACI in the discussions.

In the **Conclusions**, we have revised the reasons for the changed performance of the model results and added future perspectives for distinguishing the differences between subgrid-scale cloud microphysics and radiation feedback effects on meteorological prediction in lines 651-654 in the revised manuscript.

**2). Contradictory Model Configuration: In line 120, the authors mention that sand/dust (SD) is available in the CUACE model. However, in lines 155-156, they state, "The current scheme does not include real-time ice nucleation because dust is not available in the CUACE model." Following this, the authors apply an empirical formula for constant ice nucleation (IN). This contradictory description of the model configuration could confuse readers and should be clarified.**

Response: Revised. Thank you for the reminder. In fact, in the current model, only road dust is available and calculated in real time, while sand dust from natural sources that can act as effective ice nuclei are not available. Therefore, we clarify this in lines 155 in the revised manuscript.

**3). The ACIsub run results in more total precipitation and a higher CLWP. There**

**should be some compensating reduction in vapor at certain levels of the troposphere in the ACIsub run. However, the ACIsub simulation also shows increased vapor moisture at each vertical layer of the troposphere (Fig. 9f). It is concerning that some hydrological processes might be double-counted, or there may also be significant differences in ice-phase hydrometeors. This should be discussed in the manuscript.**

Response: Revised. According to your suggestions, we have added a discussion of the more total precipitation and higher CLWP accompanied by increased relative humidity and changed water vapor in the CU_MP_RA experiment, as well as giving ice-phase and liquid-phase hydrometeors changes in lines 513-518 in the revised manuscript, and a description of the calculations of hydrometeors processes for the cumulus convection scheme and the cloud microphysics scheme in lines 199-201 in the revised manuscript. The changes in relative humidity (RH) depend on temperature and water vapor content. As shown in Figure 9 and Figure 8, it can be seen that the RH increases in each vertical layer of the troposphere, which has a clear correspondence with the decrease in temperature. The increase in RH is more related to the decrease in temperature.

For water vapor, Figure S3 shows the comparasion of vertical profiles of vapor moisture mixing ratio between the CONTROL and CU_MP_RA experiment. It can be seen that the water vapor mixing ratio increases slightly below ~400 hPa and decreases a little above ~400 hPa. The increase in water vapor is primarily associated with the redistribution of water vapor due to the incorporation of cloud microphysics processes in the Kfeta convection scheme, as the original scheme tends to convert excessive condensed water into convective precipitation (Song and Zhang, 2011). Similar increases in water vapor accompanied by increases in precipitation can be also found in the study of Song and Zhang (2011).

[Figure]

**Figure S3: (a) The vertical profiles of vapor moisture mixing ratio in the Control and CU_MP_RA experiment. (b) The diffence of vertical vapor moisture mixing ratio between the Control and CU_MP_RA experiment. In the (a), the shadings are the spread of vapor moisture mixing ratio in six regions, and the solid lines are their average results.**

For cloud water and cloud ice content, Figure S4 shows changes in grid-scale cloud water and ice mixing ratio. The increase in total precipitation is primarily attributed to the enhancement of grid-scale precipitation. The grid-scale cloud water mixing ratio increases between ~800 hPa and ~400 hPa, while the cloud ice content also increases above approximately 500 hPa, which may be associated with enhanced convective detrainment. The increase in grid-scale precipitation is linked to the rise in both ice-phase and liquid-phase hydrometeors at grid-scale, and a detailed analysis of precipitation sources and sinks requires further discussions.

[Figure]

**Figure S4: Same with Figure S1, but for cloud water (a and b) and ice (c and d) mixing ratio at grid-scale.**

The calculations of hydromorphic processes in the cumulus convection scheme and the cloud microphysics scheme are described more detailed. We have examined in detail the calculations of hydrometeors processes in both schemes, and there is no double-counted. In the cloud microphysics scheme, we do not change the original calculation process and did not include the information of sub-grid scale clouds, and the grid-scale

hydrometeors are calculated separately. In the cumulus convection scheme, the SZ2011 scheme is directly coupled into the KFeta scheme through the one-to-one correspondence of the specific values of cloud-water mixing ratio, cloud-ice mixing ratio, precipitation production rate and snow production rate, and the subgrid-scale hydrometeors are only fed back to influence the grid-scale hydrometeors only through the detrainment and entrainment processes.

**4). The authors attribute the increased total precipitation in the ACIsub run to "Subgrid-scale ACI further enhances precipitation, especially at grid-scale." However, the results of large-scale and convective precipitation are not shown in section 5.2 of the manuscript. The partition of resolved and unresolved precipitation should be provided, at least in the supplementary material, to help readers better understand the precipitation physics in this study.**

Response: Revised. We have added the Figure S2 in the Supplement, which shows the spatial distribution of grid-scale and subgrid-scale precipitation from CONTROL, the CU-MP-RA experiment, and the difference between the CU-MP-RA and CONTROL experiment. The increase in total precipitation is mainly caused by the increase in grid-scale precipitation, which is likely related to convective detrainment of cloud ice (Song and Zhang, 2011).

[Figure]

**Figure S2. The spatial distribution of grid-scale (a-c) and subgrid-scale (d-f) precipitation from the CONTROL (left column), the CU-MP-RA experiment (middle column), and the difference between the CU-MP-RA and CONTROL experiment (right column).**

**5). The ACIsub run shows stronger relative humidity throughout the vertical atmosphere compared to the NO-ACIsub run (Fig. 9). Additionally, there is a notable increase in CLWP in the ACIsub run (Fig. 5f) compared to the NO-ACIsub run (Fig. 5e). However, the difference in cloud fraction between the ACIsub and NO-ACIsub runs is not significant. The further analysis of high, middle, and low clouds might help.**

Response: Accept. We have added Figure S1 in the Supplement, which shows the spatial distribution of high, middle, and low cloud fraction from the CONTROL and CU-MP-RA experiment, and further analyze the changes in cloud fraction with higher relative humidity and CLWP in lines 345-347 in the revised manuscript.

[Figure]

**Figure S1. The spatial distribution of high (a-b), middle (c-d), and low (e-f) cloud fraction from the CONTROL (left column) and CU-MP-RA experiment (right column).**

---

## Author Response (AR2)

Dear Editor and Reviewers:

Thank you for careful comments. These comments are all valuable and very helpful for revising and improving our paper, as well as the important guiding significance to our researches. We have made corrections which we hope meet with approval. The Reviewer 1's comments are in blue and our responses are in black. The main corrections in the paper and the point-to-point responses are as following:

**1. Regarding my review comment about "NO-ACIsub vs. ACIsub", the response and revision are acceptable, although they are different from what I asked the authors to do. However, the current title of the manuscript is slightly inconsistent with the content as a result of the revision. I suggest changing the title to highlight both the model development and the investigation.**

Response: Accept. **We have changed the title (Subgrid-scale aerosol-cloud interaction in an atmospheric chemistry model CMA_Meso5.1/CUACE and its impacts on mesoscale meteorology prediction) to highlight both the model development and the investigation**.

**2. Line 56: A period is missing..**

Response: Revised. **A period has been added in this sentence**.

**3. Lines 64-74: These sentences for the previous studies should be written in the past tense, rather than the present tense.**

Response: Revised. **We have rewritten these sentences in the past tense**.